# Near-isotropic super-resolution microscopy with axial interference speckle illumination

Hajun Yoo [1], Kwanhwi Ko [1], Sukhyeon Ka[1], Gwiyeong Moon[1,4], Hyunwoong Lee[1,5], Seongmin Im [1,6], Peng Xi [2] & Donghyun Kim [1,3] ✉

Super-resolution microscopy has pushed the limits of biological imaging. However, achieving isotropic resolution across all spatial dimensions remains a challenge and often requires a complex and highly sensitive optical setup. Herein, we introduce axial interference speckle illumination-engineered structured illumination microscopy (AXIS-SIM), a minimal-modification approach that utilizes constructive interference from a simple back-reflecting mirror to enhance the axial resolution without additional phase control or complex beam shaping. AXIS-SIM provides superior optical sectioning and improves axial resolution beyond the typical axial resolution of conventional 3D-structured illumination microscopy (~300 nm), achieving lateral and axial resolutions of 108.5 and 140.1 nm, respectively. Furthermore, its robustness against alignment errors and sample-induced aberrations enables high-throughput 3D super-resolution imaging of diverse biological specimens. We demonstrate its potential by visualizing the 3D morphology of cell membranes, resolving the nanoscale distribution of lysosomes and microtubules and tracking lysosomal movements with enhanced axial clarity.

One of the key foundations of super-resolution fluorescence microscopy, which has revolutionized biological imaging, is the spatiotemporal engineering of both the excitation light and detection signals. Various techniques, including stimulated emission depletion (STED)[1] and structured illumination microscopy (SIM)[2–5] enhance the confinement of fluorescent signals by spatially controlling excitation light. Additionally, single-molecule localization methods[6,7] and fluctuation-based imaging approaches[8,9] exploit temporally modulated fluorescent signals to surpass the diffraction limit. Each of these methods offers unique advantages for improving spatiotemporal resolution across a broad range of biological research. Despite these advancements, achieving an isotropic resolution remains a significant challenge for various super-resolution techniques[10]. With anisotropic resolution, where the lateral and axial dimensions exhibit differing levels of detail, fine structural details are often obscured along the z

axis, limiting the comprehensive understanding of three-dimensional (3D) cellular architectures.

To address the challenges of anisotropy in 3D resolution, various interference-based techniques grounded in enhanced optical sectioning that reduce out-of-focus signals have been proposed. These include sophisticated multi-objective systems[11–16], reflective mirrors[17–21], and carefully engineered interference patterns[22,23], all of which contribute to improving axial resolution by optimizing illumination on the sample. Although these approaches offer remarkable improvements in 3D imaging capabilities, they can also increase the instrumental complexity. In parallel, recent studies have explored random illumination strategies that exploit the inherent randomness of speckle patterns without prior knowledge of the illumination pattern[24–28]. These approaches simplify the generation of structured light without additional phase control or complex

[1]School of Electrical and Electronic Engineering, Yonsei University, Seoul, Korea. [2]Department of Biomedical Engineering, National Biomedical Imaging Center, College of Future TechnologyPeking University, Beijing, China. [3]Department of Biomedical Engineering, The Chinese University of Hong Kong, Shatin, N.T, Hong Kong. [4]Present address: LG Innotek, Seoul, Korea. [5]Present address: ASML Korea, Hwasung, Gyeonggi-do, Korea. [6]Present address: Nick Holonyak Micro and Nanotechnology Laboratory, University of Illinois Urbana-Champaign, Urbana, IL, USA. ✉e-mail: kimd@yonsei.ac.kr

beam shaping and provide robust performance against misalignment and sample-induced aberration.

Herein, we propose axial interference speckle-engineered SIM (AXIS-SIM), which is designed to address the challenges of both 3D resolution anisotropy and optical complexity. By employing axial interference speckle (AXIS) illumination produced through random interference in the presence of a reflective mirror, we achieved near-isotropic resolution (~150 nm) while retaining a resource-efficient optical layout. Building on the principles of super-resolution optical fluctuation imaging (SOFI)[8,29–33], our method incorporates super-resolution autocorrelation with two-step deconvolution (SACD)[34] approach to analyze temporal fluctuations under AXIS illumination. This enabled us to obtain high-resolution volumetric data with fewer raw frames, thereby reducing photobleaching and expanding the applicability of our system to biologically sensitive specimens. We highlighted the 3D visualization of cytoskeletal structures in live cells and membrane structures in fixed cells. Using AXIS-SIM, we resolved the microtubules alongside the lysosomes and presented 3D

multicolor volumetric images that captured precise positional and morphological information. We observed the membrane-bound hollow structures of lysosomes, thereby gaining a deeper insight into their structural features with high axial precision. In addition to static imaging, we took advantage of the enhanced optical sectioning of AXIS-SIM to perform time-lapse observations of lysosomes undergoing rapid intracellular movements. In this context, AXIS-SIM provides a practical solution to the long-standing challenges in 3D super-resolution microscopy with a simplified optical setup, facilitating the detailed exploration of cellular structures and dynamics in 3D.

## Results

### Experimental design and principles of AXIS-SIM for axial resolution enhancement

To experimentally validate the axial resolution enhancement achieved by AXIS-SIM, we configured an optical setup based on wide-field random speckle illumination microscopy (Fig. 1a). Similar to other methods[17,18,20,21] that employ mirrors to minimize focal spots, we placed

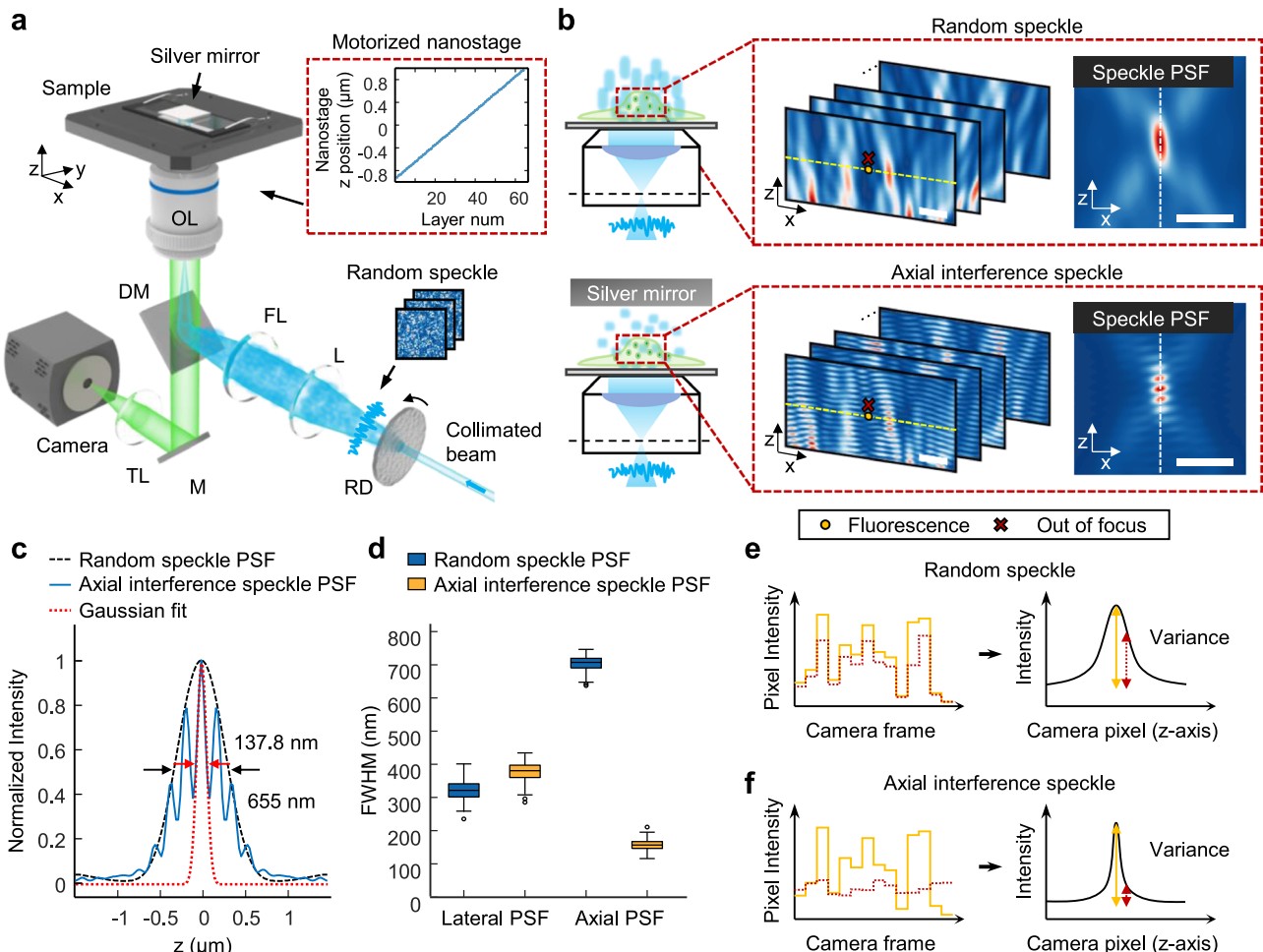

**Fig. 1 | Overview of AXIS illumination. a** Experimental optical setup of AXIS-SIM, consisting of an objective lens (OL), dichroic mirror (DM), focusing lens (FL), tube lens (TL), lens (L), rotating diffuser (RD), and mirror (M), with a silver mirror substrate placed on the sample stage. The inset illustrates the nanostage movement, capturing multiple images at each z-layer. **b** Schematic representations of beam illumination at the objective's back focal plane and sample plane: (top) random speckle illumination without axial interference, and (bottom) AXIS illumination with a reflective mirror introducing axial interference. The right insets show enlarged views of the FDTD simulations (from the red dashed boxes) and their corresponding effective speckle PSFs. The effective speckle PSF here is defined as the autocorrelation of the product of the simulated speckle illumination and the system PSF. These simulations represent a single static speckle pattern used for conceptual illustration; experimentally captured examples are shown in Supplementary Fig. 1b. **c** Line profiles corresponding to the inset in (**b**). **d** Quantification of lateral and axial PSF FWHM for captured speckle (x,z) images (*n* = 101, see Supplementary Table 1). **e**, **f** The yellow dashed horizontal lines in (b) indicate the in-focus plane where the in-focus fluorescent signal (yellow circle) is captured by the camera. Under random speckle illumination (**e**), fluorescence intensity fluctuates over time, with higher variance for the in-focus signals (yellow circle in (**b**)) compared to out-of-focus signals (red X symbols in (**b**)). In contrast, AXIS illumination (**f**) enhances optical sectioning by minimizing the effective speckle PSF size and thus outperforms random speckle illumination. Scale bars: (**b**) 1 μm.

a silver mirror substrate on the sample stage of the speckle illumination microscopy setup to generate AXIS illumination. A motorized nanostage sequentially scans along the z axis, shifting the focal depths and altering the speckle patterns at different depths. This process captures 3D images layer by layer, resulting in a full volumetric scan of the sample.

As shown in Fig. 1b, conventional random speckle illumination generates coherence-induced speckle patterns with axial elongation. In contrast, when a back-reflecting mirror is used, AXIS illumination induces constructive interference in the axial direction, leading to a significant reduction in the axial elongation of the speckle pattern. The insets in Fig. 1b show the effective speckle point spread functions (PSFs) for each condition. The speckle illumination PSF is obtained from the autocorrelation of the simulated speckle illumination alone, whereas the effective speckle PSF incorporates the influence of the detection optics by multiplying the speckle illumination by the system PSF. Notably, the effective speckle PSF of the AXIS illumination exhibited a significantly smaller full width at half maximum (FWHM) along the z-axis compared to that of the random speckle illumination (Fig. 1c).

Unlike the speckle illumination PSF derived from ideal finite-difference time-domain (FDTD) simulations (detailed in Supplementary Note 1), experimental speckle images are affected by both the excitation illumination and the detection PSF of the optical system. As a result, the experimentally observed speckle patterns are more accurately described by the effective speckle PSF, which exhibits additional z-direction sidelobes due to the detection optics (Supplementary Note 2 and Supplementary Fig. 1). Despite these factors, experimental results showed reduced axial elongation and preserved high axial spatial frequency components under AXIS illumination, driven by strong central interference in the PSF (see Supplementary Fig. 1). To further quantify this improvement, we evaluated the FWHM of the lateral and axial PSFs from multiple speckle images ($n = 101$) in the x–z plane, as shown in Fig. 1d. While the FWHM of the lateral PSF showed only a slight difference between the two methods, the FWHM of the axial effective speckle PSF in AXIS illumination was reduced by more than four-fold compared to random speckle illumination (703.5 nm/156.6 nm = 4.49).

This significantly confined axial PSF under AXIS illumination demonstrates its strong axial resolution enhancement capability, as schematically illustrated in Fig. 1e, f. In random speckle illumination, the fluorescent signal at the in-focus plane, indicated by the yellow circles in Fig. 1b, shows greater signal fluctuations over time compared to signals from out-of-focus planes (red X symbols in Fig. 1b). However, the elongated effective speckle PSF along the z-axis hinders differentiation between in-focus and out-of-focus planes (Fig. 1e). In contrast, AXIS illumination (Fig. 1f) produces a narrower axial effective speckle PSF, which causes greater variance in fluorescent signal for in-focus planes (yellow circles) compared to out-of-focus planes (red X symbols). This enhanced contrast allows clearer depth discrimination and precise optical sectioning of complex volumetric samples.

## Comparative analysis of axial interference speckle illumination

We applied AXIS-SIM to the imaging of 100 nm fluorescent beads for quantitative analysis (Fig. 2). 3D bead images were acquired by performing 62 sequential scans with a motorized nanostage spaced at 25 nm intervals. For this experiment, we allowed the fluorescent beads to settle at the bottom of the water medium and then introduced a mirror just above the sample. With a water medium, this setup helped reduce aberration compared to an air interface, while maintaining a gap of about 100 μm between the mirror and the sample to avoid direct contact. Axial distortion correction (detailed in Supplementary Note 3 and Supplementary Fig. 2) minimized residual refractive index (RI) mismatches and contributed to precise 3D imaging[35]. Although concerns may arise regarding the angle of the mirror, random speckle

patterns employed in AXIS-SIM can mitigate these potential issues. Even with the mirror tilted up to 10°, as demonstrated in the FDTD simulations (Supplementary Fig. 3), the axial resolution enhancement remained comparable to that of a perfectly aligned mirror (0°). This is a significant advantage because it allows for more flexible experimental setups without compromising the imaging quality.

For image reconstruction, we employed SACD with varying cumulant orders (detailed in Methods and Supplementary Fig. 4). The AXIS-SIM reconstruction consistently produced finer results than the diffraction-limited dynamic speckle illumination microscopy images (DL)[24], with noticeable improvements in both lateral and axial resolutions (Fig. 2a, b). Higher-order reconstructions yielded increasingly finer details, as shown in the axial cross-sectional views in Fig. 2c–e. Notably, an axial FWHM reduction of more than five-fold was observed, which indicates a significant resolution enhancement at the third order (Fig. 2f–h). The overall resolution may appear similar at first glance when comparing the presence and absence of a back-reflecting mirror (Fig. 2i–k). However, a closer look reveals that the mirror plays a crucial role in axially confining the effective speckle PSF and thereby reducing reconstruction artifacts. Without the mirror, the axial resolving power of the speckle is limited due to z-axis elongation, which poses challenges for achieving precise localization. In Fig. 2i, the red arrow indicates an artifact that did not appear in the mirrored setup (Fig. 2j). This indicates that the mirror confines the axial effective speckle PSF across the entire field and consistently improves the axial resolution. We obtained similar axial resolution enhancement in the 40 nm bead experiments, further demonstrating the consistency of AXIS-SIM performance (Supplementary Fig. 5).

## Achieving near-isotropic super-resolution in cell imaging

To further validate the capabilities of AXIS-SIM, we applied it to image ~1 μm-thick live U2OS cell microtubules stained with BioTracker 488 Green Microtubule (Fig. 3a–h and Supplementary Movie 1). The 3D cell images were acquired by performing sequential scans with a motorized nanostage spaced at 40 nm intervals along the z-axis. As the cells were maintained in phosphate-buffered saline (PBS), an axial distortion correction process was applied to each z-layer to compensate for the RI mismatch, as demonstrated in our bead experiments (see Supplementary Note 3 and Supplementary Fig. 2). AXIS-SIM provided clear depth information compared to DL and 3D Richardson-Lucy (RL)-deconvolved DL images (Fig. 3a–c). The AXIS-SIM images exhibited sharper axial views, revealing finer structural details that were blurred in the deconvolved images (Fig. 3d–f). Figure 3g, h demonstrate that AXIS-SIM can effectively resolve microtubule structures with imaging depth differences of 95.2 nm, as indicated by the white arrows. These structural changes, including the appearance and disappearance of microtubules near the white arrows, are less distinct in the RL-deconvolved images. This difference highlights the superior optical sectioning capability of the AXIS-SIM.

Next, we applied AXIS-SIM to image thicker fixed cells, specifically ~10 μm-thick U-87 MG cells labeled for membrane proteins (Fig. 3i–o and Supplementary Movie 2). The 3D cell images were acquired by performing 210 sequential scans with a motorized nanostage spaced at 40 nm intervals along the z-axis. Figure 3i compares the maximum intensity projections of U-87 MG cells imaged using DL, 3D RL-deconvolved DL, and AXIS-SIM. The AXIS-SIM image revealed enhanced resolution in a color-coded depth format, demonstrating its near-isotropic super-resolution capability. The enlarged sections (Fig. 3j, k) highlight the superior axial resolution achieved by AXIS-SIM compared to RL-deconvolved DL imaging. The axial cross-sectional views along the red dashed line in Fig. 3i highlight the superior axial resolution of AXIS-SIM (Fig. 3l). Additionally, the spatial frequency spectra presented in the log scale (Fig. 3m) and decorrelation analysis[36,37] indicate that AXIS-SIM significantly increases the spatial frequency by more than five-fold in the axial direction (788 nm/

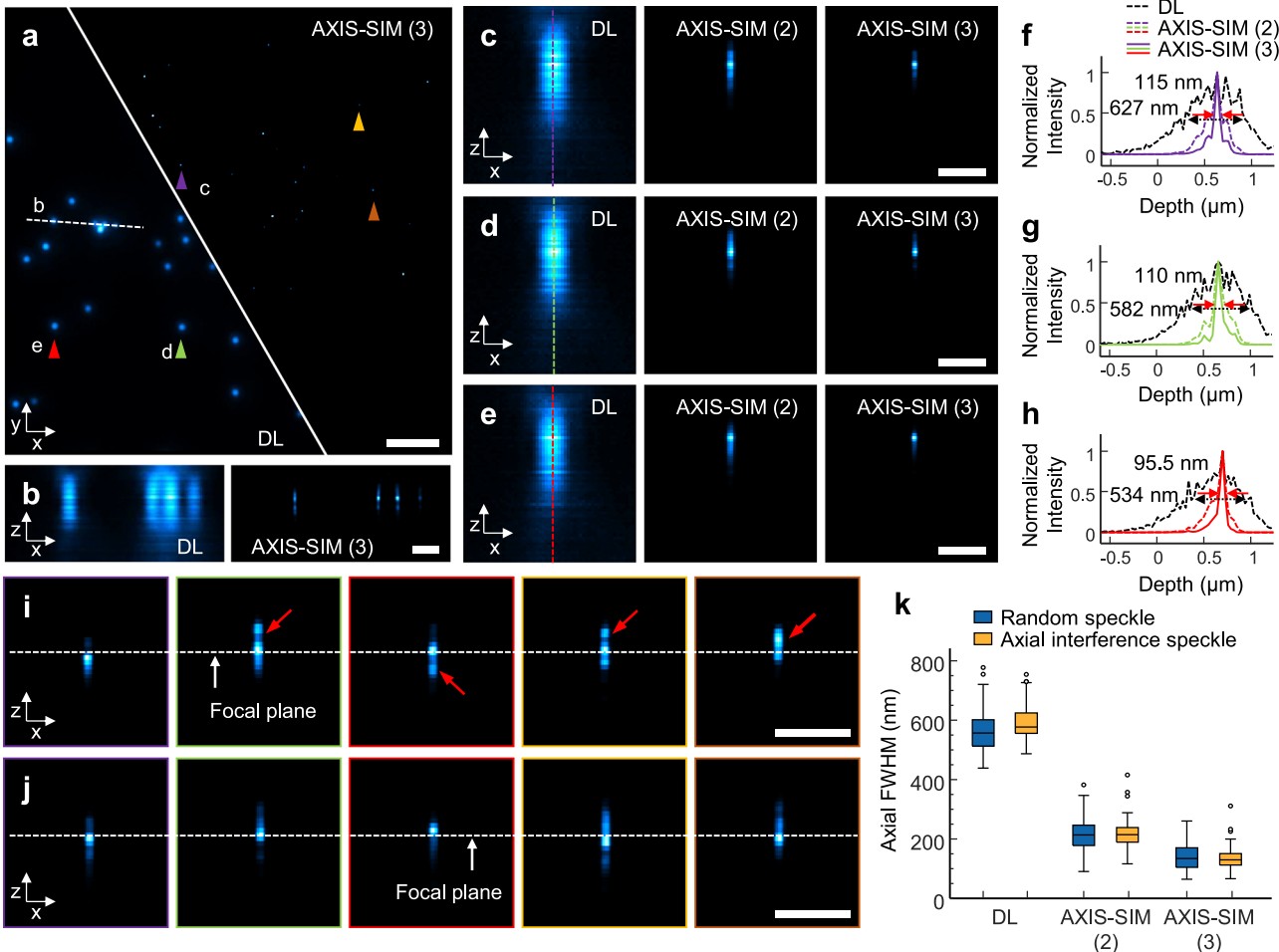

**Fig. 2 | Quantifying axial resolution improvements with AXIS-SIM. a** Maximum intensity projection of a DL image and a reconstructed image of 100-nm beads, illustrating the enhanced resolution capability of third-order AXIS-SIM. **b** Axial cross-sectional views along a dashed line in (**a**), comparing images with DL (left) and third-order AXIS-SIM (right). **c–e** Higher magnification views of a bead indicated by colored arrowheads in (**a**), demonstrating progressive improvements in axial resolution with increasing cumulant order (second-order AXIS-SIM referred to as AXIS-SIM (2), and third-order AXIS-SIM referred to as AXIS-SIM (3)). **f–h** Line profiles corresponding to the vertical dashed lines across the beads in (**c–e**), quantifying the resolution enhancement. **i, j** Axial cross-sectional images of a bead without (**i**) and with (**j**) the mirror for colored arrowheads in (**a**). The red arrow in (**i**) highlights an artifact not observed in (**j**), demonstrating the mirror's effectiveness in reducing imaging artifacts. **k** Quantification of axial FWHM measurements for beads in the DL, AXIS-SIM (2), and AXIS-SIM (3), highlighting the improved resolution at each order of AXIS-SIM ($n = 137$, see Supplementary Table 2). Scale bars: **a** 2 µm; **b–e, i, j** 500 nm.

140.1 nm = 5.62, detailed in Methods). Quantitative analysis using line profiles (Fig. 3o) along with Fig. 3n further shows the near-isotropic super-resolution characteristics of the AXIS-SIM.

To present high-throughput volumetric imaging with AXIS-SIM, we combined multiple regions of interest (ROIs) and increased the z-scan spacing to 200 nm, achieving expanded imaging areas of $86.0 \times 53.2 \times 11.7$ µm³ (Supplementary Fig. 6 and Supplementary Fig. 7). Although this approach did not achieve near-isotropic imaging due to the increased z-scan interval, it still highlights the versatility of AXIS-SIM for large-scale detailed imaging of biological systems by maintaining 3D super-resolution and expanding the field of view.

**Two-color near-isotropic super-resolution imaging with AXIS-SIM**

The use of a silver mirror in the AXIS-SIM setup proved advantageous for multicolor imaging because it allowed uniform reflection across the visible and near-infrared regions. This property, verified through FDTD calculations (Supplementary Fig. 8) and two-color bead experiments (Supplementary Fig. 9), demonstrates that the speckle illumination PSF is confined to multiple wavelengths, highlighting the

robustness of AXIS-SIM in multicolor imaging across different excitation wavelengths.

As a next step, we applied AXIS-SIM to image the microtubules and lysosomes in live U2OS cells stained with BioTracker 488 Green Microtubule and Lysotracker Deep Red. Lysosomes are membrane-bound organelles found in the cytoplasm, typically ranging in size from 100 to 500 nm, that play a critical role in cellular degradation processes and have been observed using various super-resolution microscopy[38,39]. Recent studies have shown that lysosomes move along microtubules and participate in a variety of cellular reactions as they travel[40,41]. However, it remains challenging to precisely observe their positions and movements in 3D. Compared with the RL-deconvolved DL images in 3D view, AXIS-SIM 3D images were significantly clearer in both the lateral and axial directions (Fig. 4a, b and Supplementary Movie 3). Decorrelation analysis[36] revealed noticeable improvements in lateral resolution, with AXIS-SIM achieving ~2.4–2.6-fold enhancements. Figure 4c, d show that AXIS-SIM could effectively resolve the microtubule and lysosome structures with a depth difference of 238 nm. As the imaging depths varied, distinct structural changes indicated by the white arrows became apparent, demonstrating the superior optical sectioning

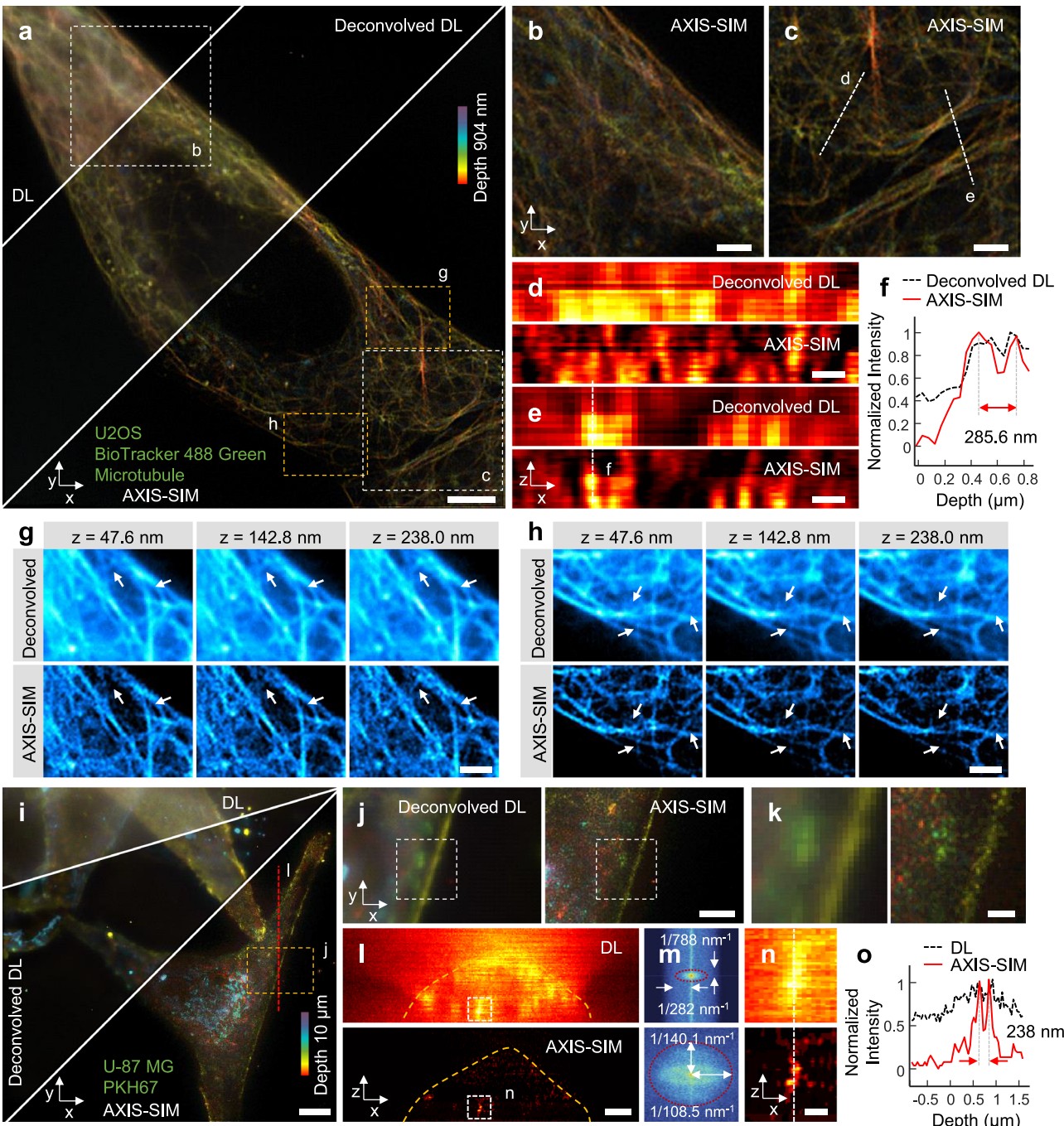

**Fig. 3 | Near-isotropic super-resolution imaging with AXIS-SIM. a** Comparison of DL, 3D RL-deconvolved results, and third-order AXIS-SIM images of live U2OS cell microtubules. The color bar indicates depth in a color-coded format. Imaging was performed over a total imaging volume of 53.2 × 53.2 × 0.9 μm³. **b, c** Enlarged views of the white dashed-box region in (**a**). **d, e** Axial cross-sectional views along the white dashed lines in (**c**), comparing the deconvolved images and the AXIS-SIM images. **f** Line profile corresponding to the vertical white dashed line in (**e**). **g, h** Enlarged views of the yellow dashed box in (**a**) showing images at z-depths of 47.6, 142.8, and 238.0 nm. The white arrow highlights changes in microtubule structure with scanning depth. These changes are less apparent in the deconvolved images, whereas AXIS-SIM clearly resolves the z-dependent sectioning of microtubules. **i** U-87 MG cell labeled for fixed membrane images using DL, 3D RL-deconvolved DL and third-order AXIS-SIM. Depth is shown using a color-coded scale. **j** Enlarged views of the yellow dashed box areas shown in (**i**) for deconvolved DL and AXIS-SIM. **k** Further enlarged version of the area within the white dashed box shown in (**j**). **l** Axial views along the red dashed line in (**i**). A yellow dashed line is included to mark the cell membrane boundary. **m** Spatial frequency spectra (in log scale) of (l). The AXIS-SIM increases the spatial frequency more than five-fold in the axial direction. **n** Further enlarged versions of the areas within the white dashed boxes in (**l**). **o** Line profiles along the white dashed line in (**n**), highlighting detailed axial resolution improvements. Scale bars: **a, i** 5 μm; **b, c, g, h, j, l** 2 μm; **d, e, k, n** 500 nm.

capability of AXIS-SIM. Additionally, axial cross-sectional views (Fig. 4e–g) and the line profile (Fig. 4h) illustrate that AXIS-SIM enables finer visualization of 3D structures, highlighting its near-isotropic super-resolution capabilities in dual-color live cell imaging.

## Live Cell Lysosome Imaging and Rapid Tracking with AXIS-SIM
In addition to mapping lysosome–microtubule interactions, we visualized the 3D distribution of lysosomes throughout the U2OS cells (Fig. 5a and Supplementary Movie 4) using AXIS-SIM. The superior optical sectioning capability of AXIS-SIM not only determined the

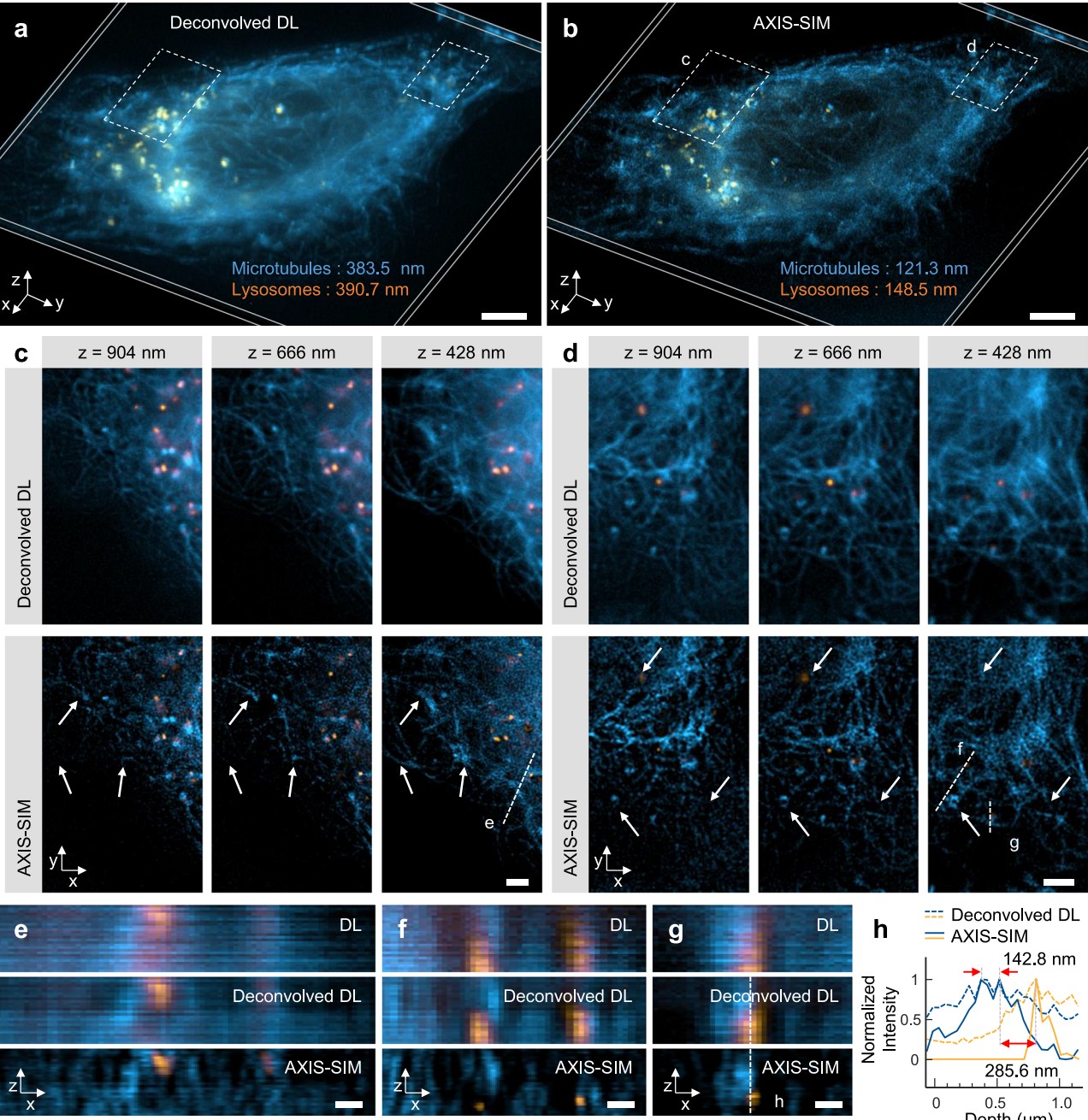

**Fig. 4 | Near-isotropic two-color imaging of live microtubules and lysosomes.**
**a, b** 3D rendered views comparing 3D RL-deconvolved and third-order AXIS-SIM images. Imaging was performed over a total volume of 53.2 × 53.2 × 1.2 μm³.
**c, d** Enlarged views of the dashed box in (**a**), showing z-slice images at depths of 904, 666, and 428 nm. White arrows indicate structural changes in microtubules and lysosomes at different scanning depth. Such changes are less apparent in the deconvolved images, whereas AXIS-SIM more effectively reveals depth-dependent sectioning, showing structures disappearing or appearing at specific planes.
**e–g** Axial cross-sectional views along the white dashed line in (**c**, **d**). **h** Line profile corresponding to the vertical white dashed line in (**g**). Scale bars: **a**, **b** 5 μm; **c**, **d** 2 μm; **e–g** 500 nm.

precise 3D spatial positions of lysosomes but also resolved the detailed 3D morphology of sufficiently large lysosomes, as shown in the enlarged views (Fig. 5b, c). Cross-sectional images in both lateral (Fig. 5d) and axial (Fig. 5e–g) views reveal the hollow structures of the lysosomes. Notably, the line profile along the white dashed line in Fig. 5e shows a hollow structure observable in the axial direction, which is invisible in the DL or RL-deconvolved DL images.

Beyond static imaging, AXIS-SIM enabled lysosomal tracking within a single focal plane (Fig. 5h–l and Supplementary Movie 5). Fig. 5h depicts lysosome distribution in a U2OS cell, whereas enlarged views (Fig. 5i, j) distinguish between stationary and dynamic lysosomes over a 30 s span. The white arrow in Fig. 5j indicates the lysosomes undergoing substantial displacement. Quantitative analysis of 15 dynamically moving lysosomes revealed an average velocity of 79.38 nm/s, and the lysosomes shown in Fig. 5j moved at 88.31 nm/s. The superior optical sectioning capability of the AXIS-SIM captured the lysosome temporarily moving out of focus, appearing smaller between 18 and 24 s before returning to the focal plane at 30 s. This high spatiotemporal resolution enables accurate tracking of rapid lysosome dynamics. Additional data on lysosomal movement are provided in Supplementary Movie 6, further demonstrating the effectiveness of AXIS-SIM in resolving and tracking intracellular processes.

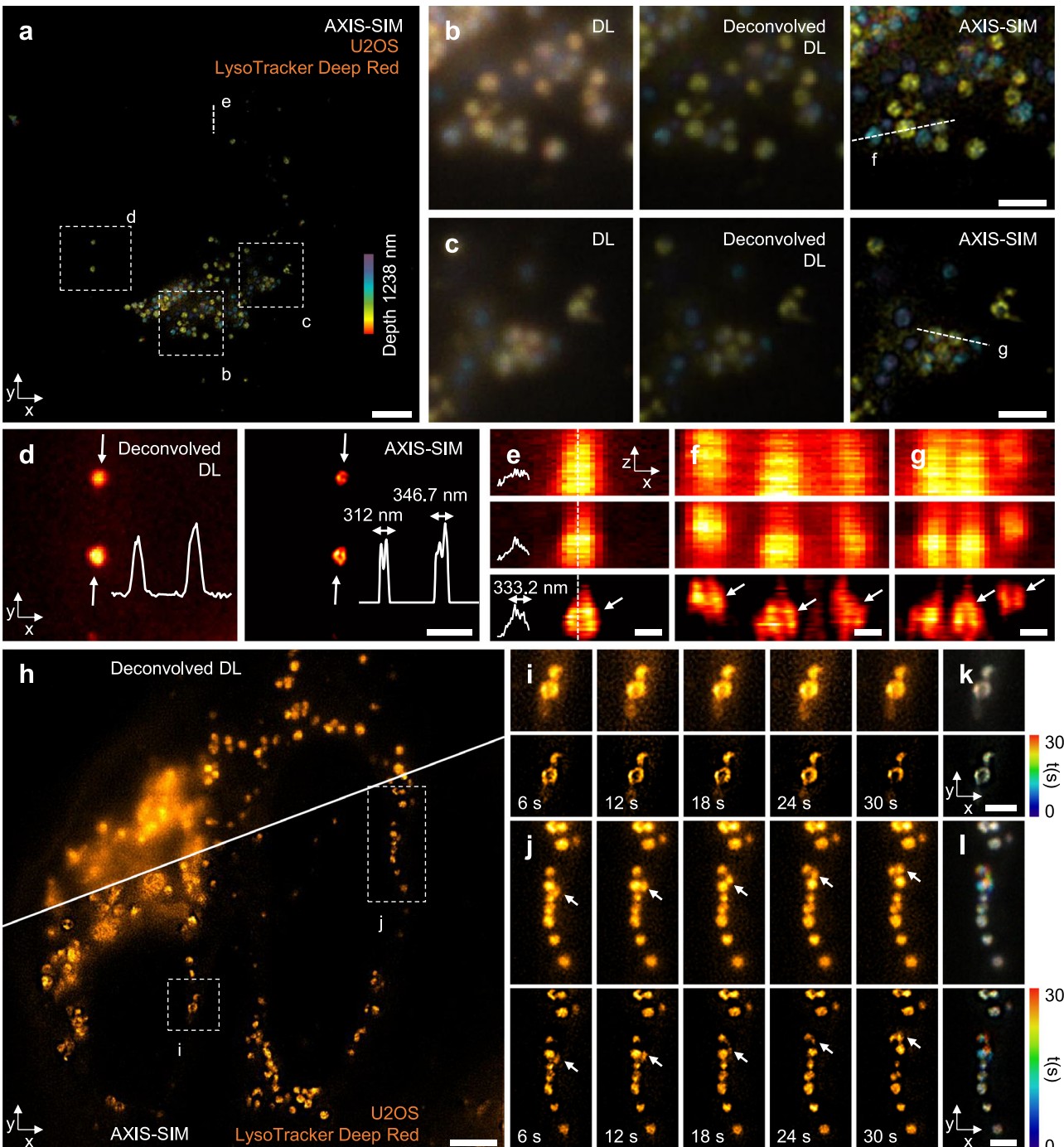

**Fig. 5 | Super-resolved 3D imaging and rapid tracking of lysosome dynamics.**
**a** Comparison of 3D RL-deconvolved and third-order AXIS-SIM images of lyso-somes in a whole U2OS cell. The color bar indicates depth in a color-coded format. Imaging was performed over a total imaging volume of $53.2 \times 53.2 \times 1.2\ \mu m^3$.
**b**, **c** Enlarged views of the region marked by the white dashed box in (**a**). **d** Enlarged view of the white dashed box in (**a**) at a depth of z = 285.6 nm, with the intensity profiles between the white arrows shown as an inset. **e–g** Axial cross-sectional views along the white dashed lines in (**a–c**), comparing DL (top), RL-deconvolved DL (middle), and AXIS-SIM images (bottom). The white arrows indicate the hollow structure of lysosomes in the axial view, with the line profile along the white dashed line shown in a left inset. **h** Lysosome distribution within a single plane of a U2OS cell. **i**, **j** Enlarged view of the region in (**h**), showing temporal changes in lysosomes tracked over time, with the white arrow indicating a lysosome undergoing sig-nificant movement. RL-deconvolved DL images (top), and the corresponding AXIS-SIM images (bottom) are displayed for comparison. **k**, **l** Color-coded representation of lysosome dynamics over a 30 s time span, showing stationary and rapid dynamics of lysosomes, respectively: (top) RL-deconvolved DL (bottom) AXIS-SIM images. Scale bars: **a**, **h** 5 μm; **b–d**, **k**, **l** 2 μm; (**e–g**) 500 nm.

## Discussion

In this study, we demonstrated the imaging capabilities of the AXIS-SIM and its advantages in addressing the challenges of 3D resolution anisotropy and optical complexity. By integrating a back-reflecting mirror into a simple random speckle imaging system, the AXIS-SIM effectively confines the axial speckle illumination PSF through con-structive interference. This confinement was achieved without the need for precise mirror alignment, highlighting the robustness of AXIS-SIM. Our experimental results across various samples were clo-sely aligned with the theoretical predictions and led to more than a

five-fold enhancement in the axial resolution compared to the diffraction-limited random speckle system. Notably, the resolution isotropy ratio (RIR), defined as the ratio of the axial to lateral resolution, quantifies the degree of resolution isotropy across different imaging techniques. AXIS-SIM achieves an RIR of 1.29 (140.1 nm/108.5 nm), positioning it within the near-isotropic resolution range while maintaining low instrumental complexity, as shown in Supplementary Fig. 10 and Supplementary Table 3.

A key strength of the AXIS-SIM is its capability to achieve near-isotropic super-resolution imaging with a relatively small number of speckle images without the need for additional optical complexity. AXIS-SIM demonstrated superior resolution using only 50–100 frames per layer (detailed in Methods), whereas conventional SOFI often requires hundreds or thousands of frames to reconstruct high-quality super-resolution image[8,31,33]. This improvement is primarily due to the incorporation of SACD[34], which significantly reduces the number of images required for SOFI reconstruction. Additionally, AXIS-SIM allows independent control over the fluctuation speed of external speckles, rather than relying solely on the natural fluctuations of fluorescent particles. Future optimizations of the speckle pattern speed and camera acquisition time could further reduce the number of images needed for reconstruction[42]. Such optimization would shorten the overall acquisition time and minimize phototoxicity, which is especially important in live-cell imaging, where prolonged light exposure can damage sensitive cellular structures.

One area for improvement in future research is to enhance the confinement of both the detection and illumination PSFs. While the AXIS illumination PSF was significantly confined along the z-axis, the detection PSF remained modestly elongated because the mirror is not positioned close enough (as noted in Supplementary Note 2 and Supplementary Fig. 1). As detailed in Supplementary Note 4 and Supplementary Fig. 11, the current mirror–sample distance ($< 100\,\mu m$) lies well within the coherence length of our illumination sources, and our calculations—as well as prior studies—indicate that interference contrast is reliably maintained up to distances of $\sim$500 $\mu m$[20]. However, further reducing the mirror height could compromise system functionality, including the wide imaging volume provided by AXIS illumination, minimal sample damage, and stable mirror alignment. To address the residual detection PSF elongation, our current approach calculates the intensity-weighting function and applies 3D deconvolution to compensate for the asymmetry of the detection PSF (Methods and Supplementary Fig. 4). Although this process slightly enhances optical sectioning, its effect remains less pronounced compared to the significant confinement of the illumination PSF. Consequently, the achieved resolution is near-isotropic rather than fully isotropic. To enhance confinement and bring the RIR closer to 1, future studies could explore strategies such as nonlinear effects of fluorescence[21]. Deep learning-based image reconstruction methods can be adopted to enhance resolution isotropy and have the potential to reduce acquisition times and minimize phototoxicity[20,43,44].

Building upon these findings, AXIS-SIM achieves substantial resolution isotropy without requiring additional phase control or complex beam shaping, an approach that, to the best of our knowledge, differs from previous SIM-based efforts aimed at enhancing axial resolution. Its compatibility with multicolor imaging and scalability for large volumetric samples make AXIS-SIM a powerful and practical tool for biological imaging. Moreover, its robustness against alignment errors and sample-induced aberrations further enhances its accessibility, enabling high-throughput 3D super-resolution imaging of diverse biological specimens.

## Methods
### Optical system design
Our custom-built AXIS-SIM system utilized an inverted microscope (Olympus, IX-73) with a ×100 oil immersion objective (NA 1.49,

Olympus, UApoN TIRF). For wide-field speckle illumination, we used a 488 nm diode laser (Coherent, OBIS LS 488 nm 100 mW) and a 633 nm HeNe laser (Melles Griot, 25-LHP-991-230) as light sources. The light was collimated using a beam expander before passing through an N-BK7 ground-glass diffuser (Thorlabs, DG10-220) mounted on a motorized precision rotation stage (Thorlabs, PRM1/MZ8) and rotated at a constant speed. The fluorescent signals were captured using an EMCCD camera (Andor, iXon Ultra 897) at 50 fps. Piezostages (Physik Instrumente, M-687 and P-545) were used to scan and capture 3D images along the z axis by sequentially varying the speckle patterns. The setup was used to image U2OS and U-87 MG cells, as well as fluorescent polystyrene beads (FluoSpheres™ Carboxylate-Modified Microspheres, $0.1\,\mu m$, 505/515 Yellow-Green and $0.2\,\mu m$, 660/680 Dark-Red, ThermoFisher Scientific). During the imaging process, we captured 50–100 images per layer to ensure spatially uniform activation of cumulative speckle-induced fluorescence across the field of view. To validate this, we compared dynamic speckle illumination (DSI) reconstructions from 50 frames with ground-truth images obtained by summing 500 frames, using structural similarity index (SSIM) as the similarity metric. The 50-frame DSI images achieved SSIM values exceeding 0.88 for 100 nm beads and 0.95 for fixed cell membranes (Supplementary Fig. 11). These results indicate that, for stationary samples such as beads and fixed cells, 50 speckle images per z-layer are sufficient to ensure a robust signal-to-noise ratio (SNR) and high-fidelity reconstruction. In contrast, live cell imaging requires 100 images per layer to compensate for the dynamic nature of organelles, such as microtubules and lysosomes, which necessitates additional frames to maintain the resolution and SNR. Preliminary tests showed that 100 frames achieved an optimal balance between capturing cellular dynamics at a high resolution and minimizing phototoxicity (Supplementary Fig. 12 and Supplementary Fig. 13). The chromatic aberration was minimized using an apochromatic objective lens designed to reduce aberrations across multiple wavelengths. To validate and correct residual aberrations, we used TetraSpeck™ Microspheres ($0.1\,\mu m$, fluorescent), imaged under 488 nm, 532 nm, and 633 nm excitation. Analysis of z-stack images revealed an axial offset below1 $\mu m$ between 488 nm and 633 nm channels. This offset was quantified and applied as a correction factor to align the z-axis data during multicolor imaging experiments to ensure proper alignment of the fluorescent signals across channels.

### Fabrication of silver mirror substrates
The silver mirrors were fabricated on BK7 glass substrates. Prior to the fabrication, the glass substrate was cleaned by sonication in acetone, isopropyl alcohol, and deionized water. A 2-nm chromium adhesion layer was first deposited onto the cleaned substrate using e-beam evaporation, followed by the deposition of a 200-nm-thick silver film via thermal evaporation. Finally, to prevent oxidation of the silver mirror, a 4.5 nm thick $Al_2O_3$ layer was formed using 50 cycles of atomic layer deposition technique. The fabricated silver mirrors were positioned on a sample with a water droplet as the intermediate layer (Fig. 1a).

### Reconstruction framework for AXIS-SIM
The reconstruction process for super-resolution imaging in AXIS-SIM largely followed the methodology used in SACD[34] (details provided in Supplementary Fig. 4). Preprocessing of the raw images and SACD reconstruction were performed using custom-developed MATLAB code, along with an ImageJ plugin for drift correction and PSF generation. The detailed steps of the reconstruction process are outlined below.

During the preprocessing phase, the sample drift was corrected using Fast4Dreg, an ImageJ plugin that aligns image stacks across time or z-planes by registering 4D data (x, y, z, time)[45]. This step is crucial for minimizing positional shifts caused by mechanical or environmental

factors, particularly those induced by the motorized nanostage that sequentially scans along the z-axis. Additionally, the 3D PSF was simulated using the Born and Wolf 3D Optical Model via the PSF Generator tool and the PSF_Generator.jar ImageJ plugin (detailed parameters are listed in Supplementary Table 4)[46]. To validate the fidelity of this theoretical model, we compared it with experimentally measured PSFs using 40 nm fluorescent beads, as shown in Supplementary Fig. 14. Following the preprocessing phase, the next step involves deconvolution using the RL algorithm, which helps enhance the image resolution and contrast[47,48]. This step refines the image by reducing the out-of-focus light and enhancing the in-focus features, thereby preparing the data for subsequent processing. As with the SACD method, this deconvolution is crucial for improving the resolution of the cumulant calculation because it filters noise, rejects out-of-focus and background signals, and enhances image clarity. To address uneven illumination and enhance signal clarity, DSI image is calculated as an intensity weighting function, $W(r)$, using the root-mean-square intensity of RL-deconvolved images over time (see Supplementary Note 2 for detailed mathematical derivation)[24]. This approach, known as quasi-confocal fluorescence sectioning, is more effective than simply summing speckle-illuminated fluorescence images to reduce background signals. When the DSI is calculated after RL deconvolution, the optical sectioning effect is significantly enhanced, enabling a more accurate extraction of information on the focal depth of interest. Furthermore, the use of AXIS illumination amplifies the impact of DSI, leading to more effective suppression of fluorescence and noise signals from defocused planes, thereby further improving image clarity and precision (as detailed in Supplementary Note 2). The DSI-based weighting function $W(r)$ is then applied to the RL-deconvolved image set using the following formula:

$$F'(r,t) = F_{RL}(r,t) \times W(r)$$

where $F_{RL}(r,t)$ represents the RL-deconvolved raw images over time. By applying this reweighting multiplication, the background noise is further suppressed, resulting in a cleaner image set that is more suitable for high-precision reconstruction. However, since $W(r)$ already contains information from the $F_{RL}(r,t)$, the fluorescence intensity is scaled to its squared value. To correct this and prevent intensity bias, we applied the square root to the $F'(r,t)$ images, thereby linearizing the brightness and compensating for the squared intensity, ensuring an accurate representation of the signal strength. After the reweighting calculation, Fourier interpolation was applied to the image stack to increase the pixel count and reduce the pixel size, while preserving the original image content[32]. Next, the autocorrelation cumulant was calculated on the Fourier-interpolated images, resulting in an enhanced resolution. This involves performing a second- or third-order cumulant calculation, similar to the method used in SOFI. By analyzing the temporal correlations of fluctuating fluorophores, the spatial resolution is further improved, with higher-order cumulants refining the image based on individual emitter fluctuations, which ultimately allows super-resolution and noise suppression[8,26]. In the final stage, 3D RL deconvolution was applied to further enhance the spatial resolution. This step, performed on the reconstructed image stack after the autocorrelation cumulant calculation, refines both the lateral and axial resolutions. Using the 3D PSF, this step corrects for any remaining blur and noise across all dimensions, effectively extending the spatial resolution in 3D. After 3D deconvolution, a final brightness linearization step was applied to suppress contrast distortions caused by non-uniform molecular brightness. Specifically, we performed an additional RL-deconvolution, followed by an $n$-th root operation (where $n$ is the cumulant order), and finally reconvolved the image with the system PSF. This procedure corrects brightness bias without compromising spatial resolution[26,31]. This process ensures that the final image retains its enhanced resolution

and contrast while minimizing the artifacts caused by unequal molecular brightness. To quantitatively optimize the deconvolution process and improve visual consistency, we introduced two additional procedures (detailed in Supplementary Note 5). We used a Fourier ring correlation (FRC)-assisted stopping criterion to objectively select the optimal number of deconvolution iterations, avoiding over-deconvolution (Supplementary Fig. 15a and Supplementary Fig. 16)[34,49]. We also applied a z-intensity flattening procedure to correct depth-dependent intensity variations and enhance image uniformity (Supplementary Fig. 15b, c). Notably, these steps primarily enhanced the visual appearance and had minimal impact on actual resolution improvement (Supplementary Table 5).

### Resolution estimation using decorrelation analysis

We estimated the spatial resolution in both the lateral (xy) and axial (z) directions using modified decorrelation analysis[36,37]. This approach determines the image resolution by identifying the local maxima of a series of decorrelation functions that correspond to the highest spatial frequency with sufficient SNR, rather than the Abbe resolution limit. For lateral resolution, analysis was performed on xy slices, and for axial resolution, a sectorial mask with a 22.5° opening angle was applied to xz slices to capture spatial frequencies along the z dimension[37]. The results provide accurate resolution estimates in both directions, showing a comparable improvement in the lateral and axial resolutions.

### Cell preparation for AXIS-SIM imaging

U-87 MG human glioblastoma cells (American Type Culture Collection, HTB-14) were cultured on glass coverslips in Dulbecco's Modified Eagle's Medium (DMEM, Sigma-Aldrich) supplemented with 10% fetal bovine serum (FBS, Gibco) and 1% penicillin-streptomycin (Gibco). The cells were incubated at 37 °C in a humidified atmosphere containing 5% $CO_2$ for 48 h. Before fixation, cells were gently rinsed with PBS to remove any residual medium. Cells were fixed with 4% formaldehyde at 4 °C for 30 min, followed by permeabilization with 0.1% Triton X-100 in PBS for 10 min. After fixation, the cell membranes were labeled using a PKH67 Green Fluorescent Cell Linker Kit (Sigma-Aldrich) according to the manufacturer's protocol. PKH67 was chosen because of its high affinity for cell membranes, which allows stable, long-term fluorescence suitable for the high-resolution imaging of membrane structures. After labeling, the samples were rinsed three times with PBS to ensure the removal of any unbound dye and prevent non-specific background fluorescence. To perform fluorescent staining of live cells, U2OS osteosarcoma cells (Korean Cell Line Bank, KCLB30096) were cultured under the same conditions as U-87 MG cells for 48 h. To stain cytoskeletal proteins, the cultured cells were rinsed with PBS to remove the culture medium and subsequently stained with Bio-Tracker™ 488 Green Microtubule Cytoskeleton Dye (Sigma-Aldrich, SCT142). The dye was diluted to a 2× concentration in complete cell culture medium containing 100 μM verapamil, followed by incubation at 37 °C for 30 min. After staining, the cells were washed with PBS and stained for lysosomes. For lysosomal staining, the probe stock solution was first diluted to 1 mM in complete cell culture medium, and LysoTracker™ Deep Red (Invitrogen, L12492) was further diluted to a final concentration of 75 nM. The cells were then incubated at 37 °C for 30 min. Similar to the U-87 MG cells, labeled cells were washed three times with PBS to remove excess dye and minimize background fluorescence.

### Data availability

Representative raw dataset required to reproduce this study's findings are available in the Figshare repository: (https://doi.org/10.6084/m9.figshare.29816543.v1)[50]. Due to the file size limitations, additional raw datasets are available from the corresponding author upon request.

## Code availability

The custom MATLAB code used for AXIS-SIM reconstruction is available in our GitHub repository: (https://github.com/AXIS-SIM/AXISSIM-reconstruction) and has been archived in Zenodo with the (https://doi.org/10.5281/zenodo.16946006[51]).

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

## Acknowledgements

This research was supported by the National Research Foundation of Korea (NRF) grant funded by the Korean government (RS-2025-00515495 and RS-2024-NR121319). H.Y. acknowledges the support by the Basic Science Research Program through the NRF funded by the Ministry of Education (RS-2023-00273526).

## Author contributions

H.Y. conceptualized the idea, performed the experiments and simulations, and developed the reconstruction framework. K.K. prepared biological samples. S.K. contributed to simulations and figure preparation. G.M., H.L. and S.I. assisted with setup design and construction. P.X. and D.K. provided early conceptual input and manuscript feedback. D.K. supervised the project. H.Y. wrote the manuscript with input and revisions from all co-authors.

## Competing interests

The authors declare no competing interests.
