## [Transparent Peer Review file · Nature Communications]

Near-isotropic Super-Resolution Microscopy with Axial Interference Speckle Illumination

Corresponding Author: Professor Donghyun Kim

Version 0:

Reviewer comments:

Reviewer #1

(Remarks to the Author)

The manuscript presents a compelling and technically rigorous advancement in the field of 3D fluorescence super-resolution microscopy. The authors introduce a method called AXIS-SIM (Axial Interference Speckle Structured Illumination Microscopy), which represents a minimalistic yet powerful modification to conventional random speckle illumination schemes. By integrating a back-reflecting silver mirror into a dynamic speckle illumination system, they achieve axial confinement of the speckle point spread function (PSF) through constructive interference, enabling enhanced axial resolution without the need for complex optical modulation, phase control, or multi-objective geometries.

The strength of the approach lies in its conceptual simplicity and optical robustness. The authors demonstrate that AXIS-SIM achieves near-isotropic super-resolution, with lateral and axial resolutions of 108.5 nm and 140.1 nm, respectively. This places the technique among the top-performing yet low-complexity 3D super-resolution methods available to date, a fact clearly evidenced by the quantitative comparisons in Supplementary Table 3 and Fig. 7. The method proves resilient against optical misalignments and sample-induced aberrations, which are often major practical hurdles in high-resolution 3D imaging. Furthermore, AXIS-SIM maintains compatibility with multicolor imaging and supports relatively fast acquisition with minimal photobleaching, making it highly applicable to both fixed and live-cell imaging scenarios.

The methodology is well described and technically sound. The theoretical foundation of AXIS illumination is clearly presented and well supported by finite-difference time-domain (FDTD) simulations and experimental speckle PSF measurements. The authors use their method for resolving intricate subcellular structures such as microtubules and lysosomes in both fixed and live U2OS cells. The ability to visualize hollow lysosomal structures in 3D and track their motion with high temporal and spatial resolution is an impressive feat of AXIS.

An interesting aspect of the work is the integration of advanced computational reconstruction methods based on SACD (super-resolution autocorrelation with two-step deconvolution), which reduces the number of required frames per z-layer.

This increases throughput and minimizes phototoxicity—an essential consideration for live imaging applications. The authors show that even with 50–100 frames per plane, AXIS-SIM outperforms conventional SOFI and deconvolved diffraction-limited imaging, achieving superior structural fidelity and optical sectioning.

In conclusion, this manuscript demonstrates a practical, scalable, and elegant solution to one of the persistent challenges in 3D fluorescence imaging: achieving near-isotropic resolution with minimal optical complexity. As such, I strongly support its publication. However, I have a few technical questions:

1. On page 11, line 283, the authors write: “During the imaging process, we captured 50–100 images per layer, ensuring sufficient time for all fluorescent signals to be fully activated.” What does “fully activated” mean? How did the authors measure/quantify that “all fluorescent signals” were “fully activated”?
2. On page 12, line 319, the authors mention that they use a Born-Wolf model for calculating a theoretical PSF that is later used for experimental image deconvolution. Did they check how well the theoretical model matches the real PSF of the optics? Often, high N.A. objectives have a slightly smaller N.A. than indicated and additional aberrations which may make the real PSF different from the ideal theoretical one.
3. On page 12, line 322m the authors mention that they used a Richardson-Lucy deconvolution. Can they specify the exact parameters? How many iterations were used in the RL deconvolution? Did they use any objective criterion telling them when to stop the RL iterations for preventing over-deconvolution (see e.g. Liu, Y., Panezai, S., Wang, Y., & Stallinga, S. (2025). Noise amplification and ill-convergence of Richardson-Lucy deconvolution. *Nature Communications*, 16(1), 911)?
4. On page 12, line 329 ff.: Can the authors present some mathematical details how exactly the weight function $W(r)$ is calculated?
5. On page 13, line 343, the authors mention that they calculate the square root of the image $F(r,t)$ for linearizing the

brightness. However, does this not also worsen the spatial resolution?

6. On page 13, line 355, one reads: "After 3D deconvolution, a final linearization step was applied to correct for variations in molecular brightness that might distort the contrast of the reconstructed images [26,31]." I recommend to include the details of this linearization step into the manuscript and not only to refer to external publications [26,31]. It is important to understand whether the linearization affects the final resolution of the image.

7. On page 14, line 403, one reads "... with the values for the water layer sourced from Palik [49] ..." Can the authors not simply provide the numerical value used for the water refractive index? And the same for the silver (and chrome?) complex-valued refractive indices?

(Remarks on code availability)

Reviewer #2

(Remarks to the Author)

Hajun Yoo et al present Axial interference speckle illumination, which achieves close to isotropic 3D resolution on the scale of 100 to 140nm. It uses a mirror to back-reflect the random laser speckles in the sample plane. The authors show 3D imaging of fluorescent nanospheres and various biological structures, including dynamics of Lysosomes in an U2OS cell.

I think overall, this is a great concept. I am usually not a fan of random speckle illumination, as it does not improve over structured illumination microscopy in terms of its spatiotemporal bandwidth, and the gain in simplicity is debatable.

In this application, however, the gain in simplicity when using speckles compared to a I5S (Shao, Lin, et al, Biophysical journal 2008) or back reflection of a laser beam in 3D SIM (Li, Xuesong, et al. Nature biotechnology 2023) is considered significant.

While I support the concept and technology, I have some concerns about the post-processing, and the artifacts shown in the biological images.

My concerns are listed below

Figure 1 b: The axial interference speckle (simulated): This is the intensity distribution of one instantaneous speckle pattern? In real life, this could not be measured, as the speckle pattern changes when a stack is acquired. Is that a correct understanding?

I find Figure 1c misleading, as to my understanding, such an axial interference PSF could not be measured, i.e. the speckle pattern would not be constant.

The experimentally measured speckle PSF (Supplementary Figure 1b and cross-section 1d) have a drastically reduced "modulation depth" compared to the simulation shown in Figure 1c. The Experimental PSF (speckle PSF times detection PSF) has very little modulation. To me, this was concerning, because this lowers the frequency support for sample-Fourier components along the z (k_z) axis. This in turn makes image restoration (deconvolution, other) numerically a harder problem. I hope the authors can comment on this.

-100nm fluorescent beads were used for the resolution characterization in Figure 2. 100nm is too big in contrast to the claimed resolution that approached 100nm (which would mean the technique had infinite resolution if the bead was already 100nm in diameter). Please repeat the measurement with smaller beads.

Can panels Figure 2i and 2j please be magnified? As it stands, visually, the improvement in axial resolution when adding the mirror is not that obvious.

Biological images in Figure 3 are somewhat artifact loaded. It is hard to tell if some features are better resolved, or if it is some speckle artifacts. In the axial views (Figure 3d and 3e) there are some notable stripe artifacts. Similar stripe artifacts are also visible in Figure 4e-f.

Overall, to me, sparser objects, like the lysosomes looked cleaner to me. This still leads me to think that the poor axial modulation of the experimental Speckle PSF makes image restoration harder for a densely labeled sample.

Further, can the authors elaborate on these points:

-What happens if the mirror is further away from the sample, will the interference contrast in the speckles reduced? Likewise, would it be generally be better if the mirror was really close to the sample? What requirements does the coherence length of the laser need to fulfill?

-The detection PSF never benefits from the mirror, I assume? So the detection PSF is just the standard widefield PSF for a single objective lens?

-The data processing involves two rounds of Richardson Lucy deconvolution, and higher order autocorrelations (cumulants) followed by linearization. The reviewer was worried that this can lead to enhancement of imaging artifacts, and that the imaging system is no longer shift invariant. Further, the multiple round of iterative deconvolution have the risk of "over-deconvolving the data". This may be apparent on the 100nm beads, where the algorithm shrinks the PSF to the size of the bead itself, or even below.

Could the authors alternatively try this algorithm: Mangeat, Thomas, et al. "Super-resolved live-cell imaging using random illumination microscopy." Cell Reports Methods 1.1 (2021).

Smaller points:

In the introduction, I would not consider "selective plane illumination microscopy" a super resolution technique.

Overall, this is a promising technology. The reviewer was however concerned about the rather weak axial modulation in the experimental PSF, and the numerous artifacts in the biological imaging.

(Remarks on code availability)

Version 1:

Reviewer comments:

Reviewer #1

(Remarks to the Author)

The authors have satisfactorily answered all my questions. I recommend now publication of the manuscript as is.

(Remarks on code availability)

The authors have satisfactorily answered all my questions. I recommend now publication of the manuscript as is.

Reviewer #2

(Remarks to the Author)

The authors have carefully addressed all my concerns. I think this was a thorough review, and the reader gets a much more detailed description of the method and its strengths and weaknesses.

There are two remaining concerns:

For the difference in modulation depth of the simulated and experimentally measured speckle PSF. The authors mention the "elongation of the detection PSF" as the cause for the difference in modulation. Can the authors modify their simulation result with a realistic detection PSF model to reproduce a similar speckle PSF as observed in Supplementary figure 1f? Does the current simulation model even involve a detection PSF?

As for "shaping the detection PSF" by bringing the mirror closer. I am not sure if that would work, as the detection PSF would change as a function of axial distance if the fluorescence light were to interfere constructively. Also, with the short coherence length (below 1 micron), interference effects that could shape the detection PSF would be limited to very shallow volumes above the coverslip.

I personally do not think this could work, so my conclusion is that the mirror placement is more governed by the coherence length of the laser and sample access considerations.

Maybe the authors can consider removing this aspect from the discussion.

(Remarks on code availability)

We sincerely thank both reviewers and the editorial office for their time and insightful feedback on our manuscript entitled “Near-isotropic Super-Resolution Microscopy with Axial Interference Speckle Illumination.” We hope and believe that their comments have greatly improved the quality and clarity of our work. We have carefully revised the manuscript point-by-point, highlighting all changes in red, and made corrections throughout the text and Supplementary Information. Below is an itemized list of these modifications in response to the reviewers’ comments.

Reviewer #1 (Remarks to the Author):

The manuscript presents a compelling and technically rigorous advancement in the field of 3D fluorescence super-resolution microscopy. The authors introduce a method called AXIS-SIM (Axial Interference Speckle Structured Illumination Microscopy), which represents a minimalistic yet powerful modification to conventional random speckle illumination schemes. By integrating a back-reflecting silver mirror into a dynamic speckle illumination system, they achieve axial confinement of the speckle point spread function (PSF) through constructive interference, enabling enhanced axial resolution without the need for complex optical modulation, phase control, or multi-objective geometries.

The strength of the approach lies in its conceptual simplicity and optical robustness. The authors demonstrate that AXIS-SIM achieves near-isotropic super-resolution, with lateral and axial resolutions of 108.5 nm and 140.1 nm, respectively. This places the technique among the top-performing yet low-complexity 3D super-resolution methods available to date, a fact clearly evidenced by the quantitative comparisons in Supplementary Table 3 and Fig. 7. The method proves resilient against optical misalignments and sample-induced aberrations, which are often major practical hurdles in high-resolution 3D imaging. Furthermore, AXIS-SIM maintains compatibility with multicolor imaging and supports relatively fast acquisition with minimal photobleaching, making it highly applicable to both fixed and live-cell imaging scenarios.

The methodology is well described and technically sound. The theoretical foundation of AXIS illumination is clearly presented and well supported by finite-difference time-domain (FDTD) simulations and experimental speckle PSF measurements. The authors use their method for resolving intricate subcellular structures such as microtubules and lysosomes in both fixed and live U2OS cells. The ability to visualize hollow lysosomal structures in 3D and track their motion with high temporal and spatial resolution is an impressive feat of AXIS.

An interesting aspect of the work is the integration of advanced computational reconstruction

methods based on SACD (super-resolution autocorrelation with two-step deconvolution), which reduces the number of required frames per z-layer. This increases throughput and minimizes phototoxicity—an essential consideration for live imaging applications. The authors show that even with 50–100 frames per plane, AXIS-SIM outperforms conventional SOFI and deconvolved diffraction-limited imaging, achieving superior structural fidelity and optical sectioning.

In conclusion, this manuscript demonstrates a practical, scalable, and elegant solution to one of the persistent challenges in 3D fluorescence imaging: achieving near-isotropic resolution with minimal optical complexity. As such, I strongly support its publication. However, I have a few technical questions:

Response: We sincerely thank the reviewer for the detailed summary and encouraging evaluation of our work.

Comment #1: *On page 11, line 283, the authors write: “During the imaging process, we captured 50–100 images per layer, ensuring sufficient time for all fluorescent signals to be fully activated.” What does “fully activated” mean? How did the authors measure/quantify that “all fluorescent signals” were “fully activated”?*

Response: We thank the reviewer for raising this important point. In our speckle-illumination scheme, “fully activated” indicates that the cumulative fluorescence response under temporal speckle modulation has reached a steady, spatially uniform level across the entire field of view—functionally equivalent to conventional wide-field excitation. To quantitatively verify this condition, we compared DSI reconstructions obtained from only 50 raw frames with a ground-truth image generated by summing 500 frames, using the structural similarity index (SSIM) as an objective metric (Supplementary Fig. 9). Even with just 50 frames, the resulting DSI images showed SSIM values above 0.88 for 100 nm beads and 0.95 for fixed U-87 MG cell membranes relative to the ground truth, indicating that key structural information was preserved and that speckle illumination had adequately covered the entire sample. In the case of live U2OS cells, we increased the number of frames to 100 per layer to account for the dynamic behavior of intracellular organelles such as microtubules and lysosomes. This ensured both high SSIM values and sufficient SNR while minimizing phototoxicity.

Revision: To improve clarity based on this comment, we have revised the paragraph on page 13 as follows:

“During the imaging process, we captured 50–100 images per layer to ensure spatially uniform activation of cumulative speckle-induced fluorescence across the field of view. To validate this, we compared DSI reconstructions from 50 frames with ground-truth images obtained by summing 500 frames, using structural similarity index (SSIM) as the similarity metric. The 50-frame DSI images achieved SSIM values exceeding 0.88 for 100 nm beads and 0.95 for fixed cell membranes (Supplementary Fig. 9). These results indicate that, for stationary samples such as beads and fixed cells, 50 speckle images per z-layer are sufficient to ensure a robust signal-to-noise ratio (SNR) and high-fidelity reconstruction. In contrast, live cell imaging requires 100 images per layer to compensate for the dynamic nature of organelles, such as microtubules and lysosomes, which necessitates additional frames to maintain the resolution and SNR. Preliminary tests showed that 100 frames achieved an optimal balance between capturing cellular dynamics at a high resolution and minimizing phototoxicity (Supplementary Fig. 9 and Supplementary Fig. 10).”

Comment #2: *On page 12, line 319, the authors mention that they use a Born-Wolf model for calculating a theoretical PSF that is later used for experimental image deconvolution. Did they check how well the theoretical model matches the real PSF of the optics? Often, high N.A. objectives have a slightly smaller N.A. than indicated and additional aberrations which may make the real PSF different from the ideal theoretical one.*

Response and revision: We thank the reviewer for raising this important point. To evaluate how well our theoretical PSF model (based on the Born-Wolf formulation with an oil-immersion objective, NA = 1.49) matches the real optical system, we performed a direct comparison using 40 nm fluorescent beads. We measured the experimental PSF from isolated beads and compared it to the theoretical PSF under the same refractive index and wavelength conditions.

As shown in the below Figure (Supplementary Fig. 11), the experimentally measured PSFs (from 40 nm fluorescent beads) and the theoretical Born–Wolf PSF exhibit a high degree of similarity in both lateral and axial views. The measured full-width at half-maximum (FWHM) values were 243.3 nm laterally and 571.8 nm axially, compared to 178.4 nm and 454.6 nm from

the theoretical model. Despite the slightly broader profiles and modest asymmetries in the experimental data, the core features of the PSF—such as lateral width and axial elongation—are well preserved. We therefore conclude that the Born–Wolf PSF provides a sufficiently accurate approximation for deconvolution in our system, balancing model fidelity with practical image reconstruction performance.

Supplementary Fig. 11 Comparison of the theoretical PSF generated using the Born–Wolf model and the experimentally measured PSF from 40 nm fluorescent beads. Normalized lateral and axial line profiles showing FWHM values of 243.3 nm vs. 178.4 nm (lateral) and 571.8 nm vs. 454.6 nm (axial) for experimental and theoretical PSFs, respectively. Despite the slightly broader profiles and modest asymmetries in the experimental data, the core features of the PSF—such as lateral width and axial elongation—are well preserved, supporting the use of theoretical PSF for deconvolution.

This validation is presented in Supplementary Fig. 11 and is now described in the revised Methods section on page 12 as follows:

“Additionally, the 3D PSF was simulated using the Born and Wolf 3D Optical Model via the PSF Generator tool and the PSF_Generator.jar ImageJ plugin (detailed parameters are listed in Supplementary Table 4)⁴⁶. To validate the fidelity of this theoretical model, we compared it with experimentally measured PSFs using 40 nm fluorescent beads, as shown in Supplementary Fig. 11. Following the preprocessing phase, ...”

Comment #3: *On page 12, line 322m the authors mention that they used a Richardson-Lucy deconvolution. Can they specify the exact parameters? How many iterations were used in the RL deconvolution? Did they use any objective criterion telling them when to stop the RL iterations for preventing over-deconvolution (see e.g. Liu, Y., Panezai, S., Wang, Y., & Stallinga, S. (2025). Noise amplification and ill-convergence of Richardson-Lucy deconvolution. Nature Communications, 16(1), 911)?*

Response: We thank the reviewer for raising this important point and for directing us to the recent work by Liu et al. (Nature Communications, 2025). We fully agree that over-deconvolution in RL can lead to noise amplification and convergence issues, particularly when the number of iterations is not well controlled. In our original analysis, we used the ‘deconvlucy’ function in MATLAB with a fixed iteration number—7 iterations for pre-deconvolution and 8 for post-deconvolution—without the use of damping or weighting parameters. These values were chosen conservatively based on preliminary visual quality assessment and literature precedent to avoid over-deconvolution. However, we recognize that such a fixed setting is not optimal for all datasets. Motivated by the reviewer’s comment, we adopted an objective stopping criterion based on Fourier Ring Correlation (FRC) resolution estimation. FRC provides a resolution metric by comparing spatial-frequency content between two statistically independent images and has been employed in determining the iteration of deconvolution by estimating a reliable cutoff frequency. For each iteration, we generated two independent subsets from the odd- and even-indexed slices of the reconstructed z-stack. Their maximum intensity projections were used to compute the 2D FRC curve, and the spatial frequency was recorded at the point where the curve crossed the 1/7 threshold. While this provides a useful approximation of lateral resolution trends, the values reflect projected—not full volumetric—resolution. We then stopped at the iteration where the FRC-estimated resolution first saturated and before any high-frequency “noise blow-up” appeared. This strategy adaptively determines the iteration number without introducing high-frequency noise amplification. It was previously suggested in Zhao et al. (Nature Photonics 17, 806–813, 2023) and line with the recommendations of Liu et al. (Nature Communications 16, 911, 2025).

We applied this FRC-assisted stopping algorithm in the post-deconvolution process of our microtubule dataset and complete iteration-versus-FRC curves, the selected stopping points, and representative FRC curve (with the 1/7 threshold) are presented in Extended Data Fig. 3a. Applying this protocol, we found that the optimal number of post-deconvolution iterations for the microtubule data was 9 (Extended Data Fig. 3a). We deeply appreciate the reviewer’s

suggestion, which directly led to a substantial improvement in our image reconstruction methodology and manuscript quality.

Extended Data Fig. 3a Enhancing AXIS-SIM with FRC-assisted deconvolution and z-intensity flattening. (a) FRC-assisted AXIS-SIM reconstruction of microtubules. Top row: diffraction-limited (DL) image and AXIS-SIM reconstructions obtained with different RL iteration numbers (9, 20, and 30). Insets show the corresponding 2D Fourier spectra. Bottom left: FRC-estimated resolution as a function of RL iteration; the red dot marks the objectively selected iteration (iteration = 9) before over-deconvolution sets in. Bottom middle/right: FRC curves (solid lines) for iteration = 9, 20, and 30, together with the 1/7 FRC threshold. The FRC-derived resolutions were 103.2 nm (iteration = 9), 87.2 nm (iteration = 20), and 79.6 nm (iteration = 30), but the latter two exhibit evident high-frequency noise blow-up and were therefore rejected by our stopping criterion.

Revision: In response to the reviewer’s comment, we have added Extended Data Fig. 3a and revised the Methods section accordingly. Specifically, we introduced a new subsection titled “FRC-assisted AXIS-SIM and z-intensity flattening” as follows, which now includes the detailed procedure used for FRC-resolution estimation and determining the optimal number of iterations.

“In our original workflow, RL-deconvolution was applied with fixed iteration numbers—7 for

pre-deconvolution and 8 for post-deconvolution—selected conservatively based on preliminary visual assessment and literature precedent to avoid over-deconvolution. However, to objectively determine the optimal number of iterations during RL-deconvolution, we then implemented a Fourier Ring Correlation (FRC)-assisted stopping criterion. FRC provides a resolution metric by comparing spatial-frequency content between two statistically independent images and has been employed in determining the iteration of deconvolution by estimating a reliable cutoff frequency³⁴. As reported, excessive iterations in RL-deconvolution can lead to a high-frequency noise amplification (so-called noise blow-up) without delivering further resolution gains⁴⁹. Accordingly, we kept the pre-deconvolution iteration fixed at an early stage (seven iterations) and applied our FRC-based stopping criterion only during post-deconvolution to select its optimal iteration number. For each iteration, the reconstructed z-stack was split into odd- and even-indexed frames, which were then merged to produce two independent subsets. Maximum intensity projections of these subsets were used to compute 2D FRC curves, and the spatial frequency at which the curve crossed the 1/7 threshold was recorded. While this approach provides an effective approximation for evaluating lateral resolution trends, it does not capture the full volumetric resolution. Because the analysis is based on 2D projections rather than the 3D distribution of signal and noise, the resulting FRC values should be interpreted as a projected-resolution estimate rather than a true 3D resolution measurement. As shown in Extended Data Fig. 3a, while the nominal FRC resolution improved with increasing iterations (e.g., 103.2 nm at 9 iterations, 87.2 nm at 20, and 79.6 nm at 30), we also observed the noise blow-up effect beyond a certain point. We therefore stopped at the iteration where the FRC-estimated resolution first saturated and before any noise amplification appeared. Applying this protocol, we found that the optimal number of post-deconvolution iterations was 9 for the microtubule data (Extended Data Fig. 3a) and 3 for the bead dataset (Supplementary Fig. 12).”

Comment #4: *On page 12, line 329 ff.: Can the authors present some mathematical details how exactly the weight function $W(r)$ is calculated?*

Response and Revision: We thank the reviewer for pointing this out. We agree that the calculation of the intensity weighting function $W(r)$ is central to the AXIS-SIM reconstruction and deserves a more explicit explanation. In the revised manuscript, we have clarified this

point by adding a reference to Supplementary Note 1, where the full derivation is presented (see Eq. 1–8, in particular Eq. 8 for the RMS expression).

$$RMS(r_d, z_c) = \rho \langle I_s^{3D} \rangle \sqrt{\int R_{RL}^{3D}(\Delta r, 0) h_{ill}^{3D}(\Delta r, 0) dr^2} \quad (\text{Eq. 8})$$

The derived RMS expression is proportional to the fluorophore distribution $\rho(r)$. Based on this result, we adopt the RMS value as the weighting function $W(r)$, which therefore carries information about the underlying fluorophore distribution. We also emphasize that this RMS-based weighting function plays a key role in enhancing fluctuation signals and suppressing out-of-focus background during cumulant processing.

To improve clarity, we now explicitly refer to Supplementary Note 1 in the main text on page 12 as follows:

“DSI image is calculated as an intensity weighting function, $W(r)$, using the root-mean-square intensity of RL-deconvolved images over time (see Supplementary Note 1 for detailed mathematical derivation)²⁴.”

Comment #5: *On page 13, line 343, the authors mention that they calculate the square root of the image $F(r,t)$ for linearizing the brightness. However, does this not also worsen the spatial resolution?*

Response: We thank the reviewer for this insightful comment. As noted, we apply a square root transform to the image stack $F'(r,t)$ before autocorrelation cumulant analysis. This step is motivated by the signal formation model in fluorescence imaging, where the observed image results from the convolution of the product of the fluorophore distribution $\rho(r)$ and the illumination $I(r,t)$ with the system PSF $h(r)$. Accordingly, Eq. 2 and Eq. 3 in Supplementary Note 1 can be simplified as:

$$F(r,t) = [\rho(r) \cdot I_{speckle}(r,t)] * h(r) \quad (\text{Eq. 2})$$

$$F_{RL}(r,t) = [\rho(r) \cdot I_{speckle}(r,t)] * h_{RL}(r) \quad (\text{Eq. 3})$$

In the context of DSI, the focal fluorescent signal $W(r)$ is modeled as the rms signal over time and is therefore theoretically proportional to $\rho(r)$ as described in Eq.8 in Supplementary Note 1.

$$RMS(r_d, z_c) = \rho \langle I_s^{3D} \rangle \sqrt{\int R_{RL}^{3D}(\Delta r, 0) h_{ill}^{3D}(\Delta r, 0) dr^2} \quad (\text{Eq. 8})$$

As a result, the image stack $F'(r, t) = F_{RL}(r, t) \times W(r)$ contains components proportional to both $\rho(r)$ and $W(r) \sim \rho(r)$, yielding an effective signal approximately proportional to $\rho^2(r)$. To avoid biasing the cumulant statistics due to higher-order fluctuation components, we apply a square root as a form of brightness linearization. This step is essential to prevent the overemphasis of central bright regions and the suppression of boundary features, which would otherwise distort the spatial distribution of cumulant-based information. This process is well established in previous SOFI-based methods and has been shown to reduce brightness skew without compromising spatial resolution [1,2]. Since resolution enhancement in our approach primarily arises during the cumulant analysis and deconvolution stages, this preprocessing step does not limit the achievable resolution.

Comment #6: *On page 13, line 355, one reads: “After 3D deconvolution, a final linearization step was applied to correct for variations in molecular brightness that might distort the contrast of the reconstructed images [26,31].” I recommend to include the details of this linearization step into the manuscript and not only to refer to external publications [26,31]. It is important to understand whether the linearization affects the final resolution of the image.*

Response and Revision: We thank the reviewer for raising this important point. In the revised manuscript, we have now included a detailed description of the linearization process rather than referring only to previous studies. As noted, the SOFI-based cumulant reconstruction may amplify molecular brightness non-uniformities, which can distort contrast in the final image. To correct this, we adopted a three-step post-processing procedure inspired by prior work [1,2], adapted to our AXIS-SIM pipeline:

The reconstructed image after cumulant analysis and post-deconvolution is further deconvolved using an RL-deconvolution algorithm. The resulting image is linearized by taking the n -th root, where n corresponds to the cumulant order. The linearized image is then reconvolved with the system PSF to restore a physically interpretable appearance. This correction helps suppress brightness-dependent contrast distortions while preserving the resolution improvement achieved by the cumulant process. Notably, this linearization step does not affect the spatial frequency content of the image or compromise resolution, as also

reported in previous SOFI-based studies [1,2]. We have revised the sentence on page 13 to reflect this process more explicitly as follows:

“After 3D deconvolution, a final brightness linearization step was applied to suppress contrast distortions caused by non-uniform molecular brightness. Specifically, we performed an additional RL-deconvolution, followed by an n -th root operation (where n is the cumulant order), and finally reconvolved the image with the system PSF. This procedure corrects brightness bias without compromising spatial resolution^{26,31}.”

Comment #7: *On page 14, line 403, one reads “... with the values for the water layer sourced from Palik [49] ...” Can the authors not simply provide the numerical value used for the water refractive index? And the same for the silver (and chrome?) complex-valued refractive indices?*

Response and revision: We thank the reviewer for this helpful suggestion. To improve clarity, we have now specified the actual refractive index values used in our simulations. Accordingly, the sentence on page 14 has been revised as “**The RI value used in the simulation was 1.515 for both the immersion oil and glass substrates, with the values for the water layer sourced from Palik⁵⁰ ($n = 1.340$ at $\lambda = 488$ nm, 1.337 at $\lambda = 532$ nm, and 1.332 at $\lambda = 633$ nm) and the complex refractive indices for the silver mirror from Johnson and Christy⁵¹ ($n + ki = 0.050 + 3.024i$ at $\lambda = 488$ nm, $0.054 + 3.434i$ at $\lambda = 532$ nm, and $0.056 + 4.285i$ at $\lambda = 633$ nm).**”

Reviewer #2 (Remarks to the Author):

Hajun Yoo et al present Axial interference speckle illumination, which achieves close to isotropic 3D resolution on the scale of 100 to 140nm. It uses a mirror to back-reflect the random laser speckles in the sample plane. The authors show 3D imaging of fluorescent nanospheres and various biological structures, including dynamics of Lysosomes in an U2OS cell.

I think overall, this is a great concept. I am usually not a fan of random speckle illumination, as it does not improve over structured illumination microscopy in terms of its spatiotemporal bandwidth, and the gain in simplicity is debatable.

In this application, however, the gain in simplicity when using speckles compared to a I5S (Shao, Lin, et al, Biophysical journal 2008) or back reflection of a laser beam in 3D SIM (Li, Xuesong, et al. Nature biotechnology 2023) is considered significant.

While I support the concept and technology, I have some concerns about the post-processing, and the artifacts shown in the biological images.

Response: Thank you for acknowledging the conceptual value and practical simplicity of AXIS-SIM. We appreciate your constructive feedback on the post-processing and will carefully address the noted artifacts and related concerns in our revised manuscript.

My concerns are listed below

Comment #1: *Figure 1 b: The axial interference speckle (simulated): This is the intensity distribution of one instantaneous speckle pattern? In real life, this could not be measured, as the speckle pattern changes when a stack is acquired. Is that a correct understanding?*

I find Figure 1c misleading, as to my understanding, such an axial interference PSF could not be measured, i.e. the speckle pattern would not be constant.

Response: We appreciate the reviewer's careful reading. Under routine imaging conditions, we acquire speckle-induced fluorescence image stacks by continuously rotating the diffuser to generate a rapidly varying speckle pattern, which results in effectively wide-field excitation when time-averaged. To illustrate the underlying principle of axial interference speckle and its point-spread function (PSF), Fig. 1b and Fig. 1c present a single static speckle realization, assuming the diffuser is held stationary. Specifically, Fig. 1b shows the FDTD simulated axial intensity distribution of this static speckle field, while Fig. 1c presents its axial PSF profile

computed as the 2D autocorrelation of the axial speckle intensity distribution. Importantly, such speckle patterns are not merely theoretical. By temporarily halting the diffuser during image acquisition, we can experimentally capture single speckle patterns under static conditions. Supplementary Fig. 1b provides representative examples of these experimentally measured speckle fields.

Revision: To prevent misinterpretation, we have revised the figure legend to clarify that Fig. 1b represents a simulated snapshot of a single speckle pattern, and that Fig. 1c illustrates the axial confinement resulting from that pattern's autocorrelation profile.

On Fig 1b. legend,

"(b) Schematic representations of beam illumination at the objective's back focal plane and sample plane: (top) random speckle illumination without axial interference, and (bottom) AXIS illumination with a reflective mirror introducing axial interference. The right insets show the FDTD simulations of the enlarged areas (red dashed boxes) and the speckle illumination autocorrelation function for each case, referred to as the speckle PSF. **These simulations represent a single static speckle pattern used for conceptual illustration; experimentally captured examples are shown in Supplementary Fig. 1b.**"

On Supplementary Fig. 1b legend,

"(b) Experimentally captured speckle images (x,z) obtained from random speckle illumination (left) and AXIS illumination (right). These images of the incident light were acquired by replacing the dichroic mirror, which is used for fluorescence signal detection, with a beam splitter. The right insets in each case represent the speckle PSF. **To acquire these speckle patterns, the diffuser rotation was temporarily halted during image acquisition, thereby reproducing the static condition assumed in the FDTD simulations.**"

Comment #2: *The experimentally measured speckle PSF (Supplementary Figure 1b and cross-section 1d) have a drastically reduced "modulation depth" compared to the simulation shown in Figure 1c. The Experimental PSF (speckle PSF times detection PSF) has very little modulation. To me, this was concerning, because this lowers the frequency support for sample-Fourier components along the z (k_z) axis. This in turn makes image restoration (deconvolution, other) numerically a harder problem. I hope the authors can comment on this.*

Response: We sincerely thank the reviewer for the insightful comment regarding the reduced modulation depth of the experimentally measured speckle PSF and its impact on axial frequency support and image restoration. The observed reduction in modulation depth (Supplementary Fig. 1b, 1f, indicated by yellow arrows) compared to the simulation (Supplementary Fig. 1e) primarily results from the elongation of the detection PSF in the z -direction, due to the mirror being positioned at a moderate height. As discussed in Supplementary Note 1 and the main text, page 9, this was an intentional design choice to preserve an imaging volume, minimize sample damage, and maintain stable mirror alignment. We recognize the reviewer’s concern that reduced modulation depth may limit axial (k_z) frequency support. In response to the reviewer’s concern, we have now included the Fourier domain representation of the speckle illumination as below figure (see Response Fig. 1 and updated Supplementary Fig. 1). As shown, AXIS illumination produces distinct high-spatial-frequency features (indicated by white arrows), which demonstrate the presence of sufficient k_z frequency content. These components are absent or strongly attenuated in random speckle illumination. The experimental FFT profile thus supports our claim that AXIS illumination enhances axial frequency support. Even though the overall modulation depth is reduced due to detection PSF elongation, the retained axial frequency bands remain sufficient for stable image restoration.

Response Fig. 1 Comparison of random speckle and axial interference speckle patterns in both real and Fourier domains from FDTD simulations and experimental results.

Furthermore, as already described in the main text and Supplementary Note 1, our processing pipeline is specifically designed to mitigate this effect. First, the RL deconvolution step reduces the detection PSF autocorrelation $R_{RL}^{3D}(\Delta r, 0)$, effectively sharpening the detection response. Second, AXIS illumination narrows the axial illumination PSF $h_{ii}^{3D}(\Delta r, 0)$ leading to over a

four-fold improvement in axial resolution compared to random speckle illumination (Fig. 1c, Supplementary Eq. 8). Lastly, an intensity-based weighting function is applied prior to cumulant computation, which suppresses sidelobes and enhances in-focus contributions. These combined strategies help ensure numerically stable image reconstruction despite the reduced modulation depth.

Revision: In response to the reviewer’s comment, we have clarified that AXIS illumination sufficiently supports axial frequency components of the sample in the z-direction. To highlight this, we updated Supplementary Fig. 1 to include Fourier domain visualizations (panels c and d; see below).

Supplementary Fig. 1 Comparison of FDTD simulations and experimentally captured random speckle and AXIS illumination (a) FDTD simulations of random speckle illumination (left) and AXIS illumination (right). The insets display the speckle illumination autocorrelation function for each case, referred to as the speckle PSF. (Identical to Fig. 1b in the main text.) (b) Experimentally captured speckle images (x,z) obtained from random speckle illumination (left) and AXIS illumination (right). These images of the incident light were acquired by replacing the dichroic mirror, which is used for fluorescence signal detection, with a beam splitter. The right insets in each case represents speckle PSF. **To acquire these speckle patterns, the diffuser rotation was temporarily halted during image acquisition, thereby reproducing the static condition assumed in**

the FDTD simulations. (c, d) Fourier domain representations of the speckle patterns shown in (a) and (b). The left and right panels correspond to random speckle and AXIS illumination, respectively. White arrows indicate high-spatial-frequency features that are distinctive of AXIS illumination, highlighting the presence of sufficient axial frequency content. (e, f) Line profiles corresponding to the dashed lines in (a) and (b). The experimentally captured AXIS illumination exhibits z-direction sidelobes in the line profile shown in (f) (yellow arrows), which result from the elongation of the detection PSF. Nevertheless, the AXIS illumination still shows reduced axial speckle elongation due to strong central interference in the illumination PSF. Furthermore, although the modulation depth is reduced by the elongated detection PSF, the axial frequency components preserved in (c, d) remain sufficient for effective image reconstruction. As noted in the main text, the volume of the detection PSF in this study was not excessively minimized, as reducing the mirror height too much could compromise the functionality of our system. This includes the wider imaging volume enabled by AXIS illumination, minimal sample damage, and stable mirror alignment. Scale bars: (a, b) 1 μm .

We also revised the paragraph on page 5 as follows:

“Unlike FDTD simulations, experimental speckle images include not only the PSF of the excitation illumination but also the influence of the detection PSF of an optical system. This resulted in additional z-direction sidelobes in the experimental speckle PSF (Supplementary Note 1 and Supplementary Fig. 1). Despite these factors, experimental results showed reduced axial elongation and preserved high axial spatial frequency components under AXIS illumination, driven by strong central interference in the PSF (see Supplementary Fig. 1).”

Comment #3: *-100nm fluorescent beads were used for the resolution characterization in Figure 2. 100nm is too big in contrast to the claimed resolution that approached 100nm (which would mean the technique had infinite resolution if the bead was already 100nm in diameter). Please repeat the measurement with smaller beads.*

Response: We thank the reviewer for raising this important point. We agree that 100 nm beads are not suitable for accurately defining resolution near or below 100 nm, and we would like to clarify that the original 100 nm bead experiments were intended to validate the imaging and reconstruction pipeline rather than define the resolution limit.

To directly address the reviewer’s concern, we repeated the resolution characterization using

40 nm fluorescent beads. The average axial FWHM measured from 40 nm beads was 125.4 ± 33.5 nm under AXIS-SIM, compared to 609.8 ± 73.0 nm under dynamic speckle illumination ($n = 44$). These values closely matched the 100 nm bead results, demonstrating that the observed resolution enhancement is not an artifact of bead size. The consistency of the FWHM values across bead sizes supports the robustness and validity of the AXIS-SIM resolution claims. We also note that resolution defined solely by FWHM can be misleading, especially in systems involving nonlinear or statistical reconstruction methods. In our view, the ability to resolve two closely spaced emitters is a more meaningful criterion. Nonetheless, the 40 nm bead results demonstrate that AXIS-SIM consistently achieves sub-diffraction resolution under more stringent conditions.

Revision: This validation is presented in Supplementary Fig. 5 and is now described in the main text on page 6 as follows:

“This indicates that the mirror confines the axial speckle PSF across the entire field and consistently improves the axial resolution. We obtained similar axial resolution enhancement in the 40 nm bead experiments, further demonstrating the consistency of AXIS-SIM performance (Supplementary Fig.5).”

Supplementary Fig. 5 Resolution analysis using 40 nm fluorescent beads. (a) Maximum intensity projection of a diffraction-limited DSI fluorescence image (DL) and a third-order AXIS-SIM reconstructions of 40 nm fluorescent beads (b) Axial cross-sectional views of three representative beads, corresponding to the colored arrowheads in (a). (c) Line profiles extracted along the dashed lines in (b). (d) Quantification of axial FWHM measurements for 100 nm and 40 nm beads under DL and AXIS-SIM conditions. For 40 nm beads, the average axial FWHM was 125.4 ± 33.5 nm for AXIS-SIM and 609.8 ± 73.0 nm for DL ($n = 44$). Scale bars: (a) $2 \mu\text{m}$; (b) 500 nm.

Comment #4: Can panels Figure 2i and 2j please be magnified? As it stands, visually, the improvement in axial resolution when adding the mirror is not that obvious.

Response and revision: We thank the reviewer for this valuable suggestion. To improve visual clarity, we have updated Fig. 2i and 2j with higher-magnified views of the beads (see Response Fig. 2). The revised panels more clearly reveal the axial features of the speckle fields under each illumination condition.

Response Fig. 2 Original and revised versions of Fig. 2i and 2j.

We would also like to clarify that the primary benefit of introducing the mirror lies not merely in apparent resolution enhancement, but in the axial confinement of the speckle PSF. Without the mirror, the speckle exhibits z-elongation, leading to insufficient axial confinement and subsequent artifacts in the reconstruction due to poor axial localization. In contrast, mirror-induced axial interference speckle results in a more confined speckle PSF, which suppresses such artifacts and improves reconstruction fidelity. As indicated by the red arrows in Fig. 2i, artifacts are observed in the mirror-free condition but are effectively suppressed in the mirrored setup (Fig. 2j).

To improve clarity based on this comment, we have revised the main text on page 6 as follows:

“The overall resolution may appear similar at first glance when comparing the presence and absence of a back-reflecting mirror (Fig. 2i–k). However, a closer look reveals that the mirror plays a crucial role **in axially confining the speckle PSF and thereby reducing reconstruction artifacts**. Without the mirror, ...”

Comment #5: *Biological images in Figure 3 are somewhat artifact loaded. It is hard to tell if some features are better resolved, or if it is some speckle artifacts. In the axial views (Figure 3d and 3e) there are some notable stripe artifacts. Similar stripe artifacts are also visible in Figure 4e–f.*

Overall, to me, sparser objects, like the lysosomes looked cleaner to me. This still leads me to think that the poor axial modulation of the experimental Speckle PSF makes image restoration harder for a densely labeled sample.

Response: We thank the reviewer for highlighting the stripe artifacts in the axial views of Figures 3d–e and 4e–f. These stripes arise from slight z-dependent intensity deviations in the speckle illumination—since individual speckles vary in brightness, reconstructed intensity can drift with imaging depth. As the reviewer correctly pointed out, this effect is more pronounced in densely labeled biological samples such as microtubules. To correct for this, we applied a post-reconstruction z-intensity flattening step. We extracted the mean intensity profile along z, smoothed it via a low-pass filter, and divided the full 3D volume by this profile to equalize the brightness across depths. As shown in Extended Data Fig. 3b, this procedure preserves structural detail while producing a more uniform appearance in the axial cross-sectional views. In Extended Data Fig. 3c, we present side-by-side axial ROIs from the main figure, shown before and after flattening. Importantly, the flattening process does not affect the measured

axial resolution. Using a decorrelation-based resolution analysis on five independent regions, we found:

Microtubule Cases	Axial resolution (Uncorrected, nm)	Axial resolution (z-intensity flattened, nm)	Resolution difference Δ (%)
Extended Data Fig. 3b	145.58	145.45	0.09
Fig. 3d	147.51	147.51	0.00
Fig. 3e	146.73	147.71	0.66
Fig. 4e	145.97	145.97	0.00
Fig. 4f	144.57	144.25	0.22

Supplementary Table 5 Quantitative evaluation of resolution change before and after z-intensity flattening.

All cases show $\Delta < 1\%$ (mean $\Delta \approx 0.19\%$), demonstrating that z-intensity flattening effectively removes stripes without compromising axial resolution. These results demonstrate that the observed stripes arise solely from an intensity non-uniformity issue and that our z-intensity flattening step effectively removes them without introducing artifacts or altering true resolution and structural integrity.

Extended Data Fig. 3b,c Enhancing AXIS-SIM with FRC-assisted deconvolution and z-intensity flattening. (b) z-Intensity flattening. Left: Maximum intensity projection of DL image, deconvolved DL, and AXIS-SIM (iteration = 9) after z-intensity correction. Center: Enlarged x-z cross-sections (locations marked by white arrows) showing uncorrected (top) versus z-flattened (bottom) data. Right: Mean z-intensity profiles of the original DL volume (gray) and the low-pass-filtered curve (red). A correction factor computed as the ratio at each depth between the

original and filtered profiles was applied to every z-slice to produce the final flattened images. (c) Representative examples of z-intensity flattening on main figures. Left panels: data from Fig. 3d, e before and after correction. Right panels: data from Fig. 4e, f before and after correction. White arrows indicate stripe artifacts that are substantially reduced following z-flattening.

Revision: In response to the reviewer's comment, we have added Extended Data Fig. 3b,c and Supplementary Table 5. Furthermore, we introduced a new subsection titled "FRC-assisted AXIS-SIM and z-intensity flattening" as follows, which now includes the detailed procedure of z-intensity flattening.

"To further improve image uniformity along the z-axis, we applied a z-intensity flattening procedure. Due to illumination inhomogeneities or signal attenuation in thick specimens, the mean intensity often varies across z-planes, which can introduce stripe-like artifacts in projections or depth-resolved views. To correct this, we computed the mean z-intensity profile of the DL image stack and applied a low-pass filter to extract the global z-trend. A slice-wise correction factor was then calculated as the ratio of the original to the smoothed profile and used to normalize each z-slice (Extended Data Fig. 3b). This flattening procedure improved visual consistency across depths and mitigated z-dependent artifacts, as demonstrated in representative images (Extended Data Fig. 3c). However, this process primarily enhances the visual appearance and had minimal impact on actual resolution improvement (less than 1% as quantified in Supplementary Table 5). These results indicate that the procedure mainly reduces stripe artifacts and does not alter the validity of the main dataset presented in the primary figures."

Further, can the authors elaborate on these points:

Comment #6: *-What happens if the mirror is further away from the sample, will the interference contrast in the speckles reduced? Likewise, would it be generally be better if the mirror was really close to the sample? What requirements does the coherence length of the laser need to fulfill?*

Response: We thank the reviewer for the thoughtful question regarding the impact of mirror distance on interference contrast and the role of laser coherence length. While it is true that increasing the distance between the mirror and the sample can reduce interference contrast, this effect becomes significant only when the optical path difference (OPD) exceeds the

coherence length of the illumination source. To examine this quantitatively, we performed a calculation interference contrast (visibility) as a function of mirror–sample distance, using typical coherence lengths for our laser sources—488 nm diode and 633 nm HeNe lasers. The 488 nm diode laser has a coherence length of several millimeters, and the 633 nm HeNe laser typically exceeds a few centimeters.

The optical path difference (OPD) introduced by the mirror is

$$OPD = 2nd, \quad (1)$$

where d is the mirror height above the sample and n the refractive index of the medium. For a gaussian-spectrum source with coherence length L_c , the field correlation envelope is

$$\gamma(d) = \exp\left(-\frac{(OPD)^2}{2L_c^2}\right), \quad (2)$$

and the resulting interference visibility follows

$$V(d) = \frac{2\gamma(d)}{1 + \gamma(d)^2}. \quad (3)$$

A detailed derivation of these relations is described in *Statistical Optics* (2nd ed.) by Goodman.

The results demonstrate that for diode lasers, the contrast begins to decline notably beyond approximately 500 μm, consistent with previous studies [3]. In contrast, for the HeNe laser, the coherence length allows interference to be maintained over much longer distances. In our experimental setup, the mirror is positioned less than 100 μm above the sample, separated by a thin water layer. This distance ensures that the optical path difference remains well within the coherence length for both laser wavelengths. As a result, both lasers operate comfortably within the coherence-effective range, and we observe strong axial interference patterns in practice. Importantly, placing the mirror closer to the sample is not always beneficial. Although

a shorter mirror–sample distance can improve interference contrast, it introduces several trade-offs. As discussed in the main text (page 9), excessively minimizing the mirror height reduces the usable imaging depth, increases the risk of sample damage, and compromises mirror alignment stability. In our system, the ~100 μm spacing represents a practical balance between maximizing interference efficiency and ensuring robust performance for live-cell imaging.

Revision: We have included a calculation result in Supplementary Note 3, which illustrates the relationship between mirror height, laser coherence length, and interference contrast, and clarified these points in the revised manuscript in the Discussion section as follows:

“One area for improvement in future research is to enhance the confinement of both the detection and illumination PSFs. While the AXIS illumination PSF was significantly confined along the z-axis (Fig. 2e), the detection PSF remained modestly elongated because the mirror is not positioned close enough (as noted in Supplementary Note 1 and Supplementary Fig. 1). As detailed in Supplementary Note 3, the current mirror–sample distance ($< 100 \mu\text{m}$) lies well within the coherence length of our illumination sources, and our calculations—as well as prior studies—indicate that interference contrast is reliably maintained up to distances of approximately $500 \mu\text{m}$ ²⁰. However, further reducing the mirror height could compromise system functionality, including the wide imaging volume provided by AXIS illumination, minimal sample damage, and stable mirror alignment. To address the residual detection PSF elongation, our current approach calculates the intensity-weighting function and applies 3D deconvolution to compensate for the asymmetry of the detection PSF (Methods and Supplementary Fig. 4). Although this process slightly enhances optical sectioning, its effect remains less pronounced compared to the significant confinement of the illumination PSF. Consequently, the achieved resolution is near-isotropic rather than fully isotropic. To enhance confinement and bring the RIR closer to 1, future studies could explore strategies such as positioning the mirror closer to the sample to better influence the shape of the detection PSF or using the nonlinear effects of fluorescence²¹. Deep learning-based image reconstruction methods can be adopted to enhance resolution isotropy and have the potential to reduce acquisition times and minimize phototoxicity^{20,43,44}.”

We appreciate the reviewer’s insightful comment, which helped improve the clarity and rigor of our design rationale.

Comment #7: *-The detection PSF never benefits from the mirror, I assume? So the detection PSF is just the standard widefield PSF for a single objective lens?*

Response and revision: We thank the reviewer for this important clarification. In our current implementation, the mirror is positioned approximately 100 μm above the sample—sufficiently distant from the opposing objective lens that the detection PSF itself is initially similar to a standard wide-field PSF acquired through a single high-NA objective. In practice, the overall axial resolving power is governed jointly by both the detection and illumination PSFs. While the detection PSF remains elongated, the AXIS illumination PSF is substantially compressed along the z-axis, driving a four-fold improvement in axial resolution versus random speckle (main text Fig. 1c). As a result, the final imaging performance is determined by the modulated excitation rather than being limited by the elongated detection PSF alone. Importantly, despite the elongation of the detection PSF, the experimentally captured AXIS illumination contains high-spatial-frequency components, as evidenced by the axial sidebands in the Fourier domain as below figure (see Response Fig. 1 and updated Supplementary Fig. 1, white arrows). Moreover, we further mitigate the axial extent of the detection PSF using RL deconvolution, which significantly reduces its autocorrelation width $R_{\text{RL}}^{3D}(\Delta r, 0)$, as shown in Supplementary Eq. 8. Therefore, while the detection PSF is not directly modified by mirror-induced optical effects, its influence on final image quality is effectively addressed through our reconstruction pipeline. To clarify these points, as noted in our response to Comment #2, we have updated Supplementary Fig. 1 to include the Fourier domain representation of the AXIS illumination and revised the main text accordingly to explain the presence of high axial spatial frequencies under AXIS illumination.

Supplementary Fig. 1 Comparison of FDTD simulations and experimentally captured random speckle and AXIS illumination (a) FDTD simulations of random speckle illumination (left) and AXIS illumination (right). The insets display the speckle illumination autocorrelation function for each case, referred to as the speckle PSF. (Identical to Fig. 1b in the main text.) (b) Experimentally captured speckle images (x,z) obtained from random speckle illumination (left) and AXIS illumination (right). These images of the incident light were acquired by replacing the dichroic mirror, which is used for fluorescence signal detection, with a beam splitter. The right insets in each case represents speckle PSF. To acquire these speckle patterns, the diffuser rotation was temporarily halted during image acquisition, thereby reproducing the static condition assumed in the FDTD simulations. (c, d) Fourier domain representations of the speckle patterns shown in (a) and (b). The left and right panels correspond to random speckle and AXIS illumination, respectively. White arrows indicate high-spatial-frequency features that are distinctive of AXIS illumination, highlighting the presence of sufficient axial frequency content. (e, f) Line profiles corresponding to the dashed lines in (a) and (b). The experimentally captured AXIS illumination exhibits z -direction sidelobes in the line profile shown in (f) (yellow arrows), which result from the elongation of the detection PSF. Nevertheless, the AXIS illumination still shows reduced axial speckle elongation due to strong central interference in the illumination PSF. Furthermore, although the modulation depth is reduced by the elongated detection PSF, the axial frequency components preserved in (c, d) remain sufficient for effective image reconstruction. As noted in the main text, the volume of the detection PSF in this study was not excessively minimized, as reducing the mirror height too much could compromise the functionality of our system. This includes the wider imaging volume enabled by AXIS illumination, minimal sample damage, and stable mirror alignment. Scale bars: (a, b) 1 μm .

Comment #8: *-The data processing involves two rounds of Richardson Lucy deconvolution, and higher order autocorrelations (cumulants) followed by linearization. The reviewer was worried that this can lead to enhancement of imaging artifacts, and that the imaging system is no longer shift invariant. Further, the multiple round of iterative deconvolution have the risk of “over-deconvolving the data”. This may be apparent on the 100nm beads, where the algorithm shrinks the PSF to the size of the bead itself, or even below.*

Response: We appreciate the reviewer’s concern regarding potential artifact enhancement, loss of shift invariance, and over-deconvolution arising from our dual-stage RL deconvolution combined with higher-order SOFI cumulants. To address the risk of over-deconvolution, we adopted an objective stopping criterion based on Fourier Ring Correlation (FRC) resolution estimation. Specifically, for each iteration, we generated two independent subsets from the odd- and even-indexed slices of the reconstructed z-stack. Their maximum intensity projections were used to compute the 2D FRC curve, and the spatial frequency was recorded at the point where the curve crossed the 1/7 threshold. While this provides a useful approximation of lateral resolution trends, the values reflect projected—not full volumetric—resolution. We then selected the iteration at which the FRC-estimated resolution first saturated and before the emergence of high-frequency artifacts (noise blow-up). This strategy allows adaptive determination of the optimal iteration number while avoiding overfitting and noise amplification. Our approach is inspired by the method proposed by Zhao et al. (Nature Photonics 17, 806–813, 2023) and is consistent with the recommendations outlined by Liu et al. (Nature Communications 16, 911, 2025). The full procedure and representative examples are presented in Extended Data Fig. 3a. As shown in Supplementary Fig. 12a, reprocessing the 100 nm bead data under this FRC-guided scheme (iter = 3 instead of 8) substantially alleviated lateral over-minimization and restored the expected spherical morphology, without compromising axial performance (axial FWHM remained on the order of 100–120 nm).

Regarding some results where the reconstructed FWHM drops below 100 nm, we emphasize that this does not represent an unphysical artifact. Rather, it reflects the well-known sharpening effect of higher-order SOFI cumulants, which are non-linear and mathematically narrow the effective point spread function (PSF). Higher-order (n) cumulants progressively sharpen the effective PSF, approximately scaling as $1/\sqrt{n}$ under the Gaussian approximation [4]. As a result, the intensity profile of a 100 nm bead can exhibit an apparent FWHM smaller

than its physical diameter without indicating algorithmic over-sharpening. In our data, this sharpening effect is observed consistently across 137 beads, with no evidence of negative lobes or morphological distortion. To prevent conflating object size with imaging resolution, we do not use bead FWHM as a resolution metric. Instead, we quantify final estimated resolution using decorrelation analysis. To verify that our processing remains spatially invariant, we computed tile-wise local estimated resolution maps (4x4 grid) across a single field of view (see Supplementary Fig. 12b,c). Although the absolute, decorrelation-based resolution estimates around 60 nm—reflecting the PSF sharpening—the standard deviation across tiles is just 8.9%. This tight spatial uniformity demonstrates that our pipeline applies equally at every location, confirming approximate shift-invariance.

Revision: In response to the reviewer's comment, we have added Extended Data Fig. 3a and revised the Methods section accordingly. Specifically, we introduced a new subsection titled "FRC-assisted AXIS-SIM and z-intensity flattening", which now details the procedure for FRC-based resolution estimation and for determining the optimal number of RL-deconvolution iterations. We have also included Supplementary Fig. 12, which demonstrates the application of this FRC-assisted stopping strategy to bead data.

Extended Data Fig. 3 Enhancing AXIS-SIM with FRC-assisted deconvolution and z-intensity flattening. (a) FRC-assisted AXIS-SIM reconstruction of microtubules. Top row: diffraction-limited (DL) image and AXIS-SIM reconstructions obtained with different RL iteration numbers (9, 20, and 30). Insets show the corresponding 2D Fourier spectra. Bottom left: FRC-estimated resolution as a function of RL iteration; the red dot marks the objectively selected iteration (iteration = 9) before over-deconvolution sets in. Bottom middle/right: FRC curves (solid lines) for iteration = 9, 20, and 30, together with the 1/7 FRC threshold. The FRC-derived resolutions were 103.2 nm (iteration = 9), 87.2 nm (iteration = 20), and 79.6 nm (iteration = 30), but the latter two exhibit evident high-frequency noise blow-up and were therefore rejected by our stopping criterion. (b) z-Intensity

flattening. Left: Maximum intensity projection of DL image, deconvolved DL, and AXIS-SIM (iteration = 9) after z-intensity correction. Center: Enlarged x–z cross-sections (locations marked by white arrows) showing uncorrected (top) versus z-flattened (bottom) data. Right: Mean z-intensity profiles of the original DL volume (gray) and the low-pass-filtered curve (red). A correction factor computed as the ratio at each depth between the original and filtered profiles was applied to every z-slice to produce the final flattened images. (c) Representative examples of z-intensity flattening on main figures. Left panels: data from Fig. 3d, e before and after correction. Right panels: data from Fig. 4e, f before and after correction. White arrows indicate stripe artifacts that are substantially reduced following z-flattening.

Supplementary Fig. 12 Artifact suppression and spatial uniformity in AXIS-SIM reconstruction. (a) Axial cross-sections of individual 100-nm beads acquired in DL and in second-order and third-order AXIS-SIM (denoted AXIS-SIM (2) and AXIS-SIM (3), respectively). For each order, results are shown after 3 and 8 RL-deconvolution iterations. Although the bead

FWHM does not directly represent system resolution, applying FRC-assisted iteration stopping (iter = 3) successfully alleviated lateral over-minimization while preserving axial confinement (~100–120 nm FWHM), resulting in more spherical bead morphology in 3D. The dataset AXIS-SIM (3) iter 8, corresponds to that shown in Fig. 2c–e of the main text. (b) Maximum intensity projection of the DL bead volume, subdivided into a 4 × 4 grid. Local lateral resolution in each tile was estimated using decorrelation analysis. The resulting resolution maps for DL, AXIS-SIM (2), and AXIS-SIM (3) at iter = 3 show consistent improvements with increasing order and minimal spatial variation across the field of view. (c) Histogram of local resolution values from the 13 central tiles in (b). Mean ± s.d. are indicated for each dataset. The narrow and consistent distributions suggest that the reconstruction process preserves spatial uniformity and approximate shift invariance. Scale bars: (a) 500 nm; (b) 2 μm.

Comment #9: *Could the authors alternatively try this algorithm: Mangeat, Thomas, et al. "Super-resolved live-cell imaging using random illumination microscopy." Cell Reports Methods 1.1 (2021).*

Response: We thank the reviewer for suggesting the algorithm proposed by Mangeat et al. (Cell Reports Methods, 1.1, 2021), named AlgoRIM. We are aware of its strengths and have tested it on our microtubule dataset (Fig. 3a–h, see Response Fig. 3). While it produced reasonable results, it required at least 800 raw images to achieve optimal performance, significantly more than the 50–100 frames used in our AXIS-SIM workflow. When comparing reconstruction quality in terms of axial resolution and contrast, AlgoRIM did not outperform AXIS-SIM under equivalent conditions (e.g., with 100 frames). Given these limitations and the constraints of our imaging system, we chose to adopt SACD, which is optimized for our speckle-modulated datasets and achieves robust results with only 50–100 frames.

Response Fig. 3 Comparison of AXIS-SIM and AlgoRIM reconstructions

Smaller points:

Comment #10: *In the introduction, I would not consider “selective plane illumination microscopy” a super resolution technique.*

Response and revision: We thank the reviewer for this clarification. We agree that traditional selective plane illumination microscopy (SPIM) is not categorized as a super-resolution technique. In our revised manuscript, we have removed SPIM from the list of super-resolution methods. We have also updated the citation numbering accordingly.

On Page 3,

“One of the key foundations of super-resolution fluorescence microscopy, which has revolutionized biological imaging, is the spatiotemporal engineering of both the excitation light and detection signals. Various techniques, including stimulated emission depletion (STED)¹ and structured illumination microscopy (SIM)^{2–5} enhance the confinement of fluorescent signals by spatially controlling excitation light. Additionally, single-molecule localization methods^{6,7} and fluctuation-based imaging approaches^{8,9} exploit temporally modulated fluorescent signals to surpass the diffraction limit.”

Overall, this is a promising technology. The reviewer was however concerned about the rather weak axial modulation in the experimental PSF, and the numerous artifacts in the biological imaging.

Response: We sincerely appreciate the reviewer's concise summary and positive evaluation of our manuscript.

Referencesss

1. Geissbuehler, S. et al. Mapping molecular statistics with balanced super-resolution optical fluctuation imaging (bSOFI). *Opt. Nanoscopy* **1**, 4 (2012).
2. Kim, M., Park, C., Rodriguez, C., Park, Y. & Cho, Y.-H. Superresolution imaging with optical fluctuation using speckle patterns illumination. *Sci. Rep.* **5**, 16525 (2015).
3. Li, X. et al. Three-dimensional structured illumination microscopy with enhanced axial resolution. *Nat. Biotechnol.* **41**, 1307–1319 (2023).
4. Dertinger, T., Colyer, R., Iyer, G., Weiss, S. & Enderlein, J. Fast, background-free, 3D super-resolution optical fluctuation imaging (SOFI). *Proc. Natl. Acad. Sci.* **106**, 22287–22292 (2009).

We sincerely thank both reviewers and the editorial office for their time and constructive feedback on our manuscript entitled “Near-isotropic Super-Resolution Microscopy with Axial Interference Speckle Illumination.” The thoughtful comments are highly appreciated, which we believe have further strengthened the quality and clarity of our work. In this second revision, we have addressed the remaining points raised by the reviewers, esp. the Reviewer #2, and the editorial office. All changes are marked in red in the revised manuscript and Supplementary Information. Below is an itemized list of these modifications in response to the reviewers’ comments.

Reviewer #1 (Remarks to the Author):

The authors have satisfactorily answered all my questions. I recommend now publication of the manuscript as is.

Response: We sincerely thank the reviewer for the positive assessment and recommendation for publication.

Reviewer #1 (Remarks on code availability):

The authors have satisfactorily answered all my questions. I recommend now publication of the manuscript as is.

Response: Again, we appreciate the reviewer’s confirmation that our responses have satisfactorily addressed all questions.

Reviewer #2 (Remarks to the Author):

The authors have carefully addressed all my concerns. I think this was a thorough review, and the reader gets a much more detailed description of the method and its strengths and weaknesses.

Response: We sincerely thank the reviewer for acknowledging that we have carefully addressed all concerns and that the revised manuscript now provides a more detailed description of the method and its strengths and weaknesses.

There are two remaining concerns:

Comment #1: For the difference in modulation depth of the simulated and experimentally measured speckle PSF. The authors mention the "elongation of the detection PSF" as the cause for the difference in modulation. Can the authors modify their simulation result with a realistic detection PSF model to reproduce a similar speckle PSF as observed in Supplementary figure 1f? Does the current simulation model even involve a detection PSF?

Response: We thank the reviewer for this insightful comment. The reviewer is correct that our original calculation of the speckle PSF did not include the effects of the detection optics. To address this, we revised our simulation to incorporate a realistic model of the system's detection PSF and defined the effective speckle PSF. As shown in Response Fig. 1, this updated model yields a speckle PSF whose modulation depth and overall shape closely match the experimental measurements.

For clarity, we now explicitly define these terms in the revised manuscript: (i) the speckle illumination PSF refers to the autocorrelation of the simulated speckle illumination alone, and (ii) the effective speckle PSF refers to the revised model that accounts for the detection optics and is defined as the autocorrelation of the product of the speckle illumination and the system's detection PSF. We believe this additional analysis provides a more realistic representation of the experimental conditions and directly addresses the reviewer's concern.

Response Fig. 1 Calculation of the effective speckle PSF of AXIS illumination. The ideal illumination pattern is first multiplied by the detection PSF, and its autocorrelation then yields the final effective speckle PSF of the imaging system.

Revision: The term speckle PSF has been explicitly distinguished into speckle illumination PSF and effective speckle PSF throughout the main text and the Supplementary Information. Fig. 1b, c and Supplementary Fig. 1a, e have been replaced with the newly calculated effective speckle PSF, and the corresponding main text and figure legends have been updated as follows:

Fig. 1 Overview of AXIS illumination. (a) Experimental optical setup of AXIS-SIM, consisting of an objective lens (OL), dichroic mirror (DM), focusing lens (FL), tube lens (TL), lens (L), rotating

diffuser (RD), and mirror (M), with a silver mirror substrate placed on the sample stage. The inset illustrates the nanostage movement, capturing multiple images at each z-layer. (b) Schematic representations of beam illumination at the objective's back focal plane and sample plane: (top) random speckle illumination without axial interference, and (bottom) AXIS illumination with a reflective mirror introducing axial interference. The right insets show enlarged views of the FDTD simulations (from the red dashed boxes) and their corresponding effective speckle PSFs. The effective speckle PSF here is defined as the autocorrelation of the product of the simulated speckle illumination and the system PSF.

Supplementary Fig. 1 Comparison of FDTD simulations and experimentally captured random speckle and AXIS illumination (a) FDTD simulations of random speckle illumination (left) and AXIS illumination (right). The insets display the effective speckle PSF for each case, defined as the autocorrelation of the product of the FDTD-simulated speckle illumination and the system PSF. Identical to Fig. 1b in the main text.

On main text page 5,

“The insets in Fig. 1b show the effective speckle point spread functions (PSFs) for each condition. The speckle illumination PSF is obtained from the autocorrelation of the simulated speckle illumination alone, whereas the effective speckle PSF incorporates the influence of the detection optics by multiplying the speckle illumination by the system PSF. Notably, the

effective speckle PSF of the AXIS illumination exhibited a significantly smaller full width at half maximum (FWHM) along the z-axis compared to that of the random speckle illumination (Fig. 1c).

Unlike the speckle illumination PSF derived from ideal FDTD simulations, experimental speckle images are affected by both the excitation illumination and the detection PSF of the optical system. As a result, the experimentally observed speckle patterns are more accurately described by the effective speckle PSF, which exhibits additional z-direction sidelobes due to the detection optics (Supplementary Note 1 and Supplementary Fig. 1).”

Comment #2: As for "shaping the detection PSF" by bringing the mirror closer. I am not sure if that would work, as the detection PSF would change as a function of axial distance if the fluorescence light were to interfere constructively. Also, with the short coherence length (below 1 micron), interference effects that could shape the detection PSF would be limited to very shallow volumes above the coverslip.

I personally do not think this could work, so my conclusion is that the mirror placement is more governed by the coherence length of the laser and sample access considerations.

Maybe the authors can consider removing this aspect from the discussion.

Response and revision: We thank the reviewer for this valuable point. Upon reconsideration, we agree that shaping the detection PSF by simply bringing the mirror closer presents significant practical challenges, especially considering the short coherence length and the axial-dependent interference effects as the reviewer pointed out.

We have acknowledged the reviewer suggestion and therefore removed this aspect from the discussion. The revised sentence now focuses on nonlinear effects, as follows:

“To enhance confinement and bring the RIR closer to 1, future studies can explore strategies such as nonlinear effects of fluorescence²¹.”

We are grateful to the reviewer for highlighting this point, which helped us sharpen the scope of our discussion.